# Optimizer-Dependent Generalization Bound for Quantum Neural Networks

## Abstract

Quantum neural networks (QNNs) play a pivotal role in addressing complex tasks within quantum machine learning, analogous to classical neural networks in deep learning. Ensuring consistent performance across diverse datasets is crucial for understanding and optimizing QNNs in both classical and quantum machine learning tasks, but remains a challenge as QNN's generalization properties have not been fully explored. In this paper, we investigate the generalization properties of QNNs through the lens of learning algorithm stability, circumventing the need to explore the entire hypothesis space and providing insights into how classical optimizers influence QNN performance. By establishing a connection between QNNs and quantum combs, we examine the general behaviors of QNN models from a quantum information theory perspective. Leveraging the uniform stability of the stochastic gradient descent algorithm, we propose a generalization error bound determined by the number of trainable parameters, data uploading times, dataset dimension, and classical optimizer hyperparameters. Numerical experiments validate this comprehensive understanding of QNNs and align with our theoretical conclusions. As the first exploration into understanding the generalization capability of QNNs from a unified perspective of design and training, our work offers practical insights for applying QNNs in quantum machine learning.

## 1 Introduction

Quantum computing leverages the laws of quantum mechanics to solve complex problems more efficiently than classical computers, offering notable quantum speedups in areas such as cryptography (Shor, 1997), and quantum simulations (Lloyd, 1996; Childs et al., 2018). Recent advancements in quantum hardware have demonstrated quantum advantages in specific tasks (Arute et al., 2019; Zhong et al., 2020; Wu et al., 2021), catalyzing the exploration of quantum computing's potential in artificial intelligence. This interdisciplinary connection has given rise to *quantum machine learning* (Biamonte et al., 2017; You et al., 2023; Tang & Yan, 2022; Liu et al., 2022; Caro et al., 2022b; Cerezo et al., 2022; Huang et al., 2022b; Qian et al., 2022a; Yu et al., 2022a; Tian et al., 2023; Li et al., 2019; 2022; Jerbi et al., 2023a; Li & Deng, 2022; Huang et al., 2022a).

One of the leading frameworks in quantum machine learning is the *quantum neural network* (QNN), which represents the quantum analog of classical artificial neural networks and typically refers to parameterized quantum circuits that are trainable based on quantum measurement results. A well-known architecture within QNNs is the data re-uploading QNN (Pérez-Salinas et al., 2020; Gil Vidal & Theis, 2020), which integrates multiple training and data encoding layers within a single quantum circuit. This approach significantly enhances the expressivity of the models, allowing them to approximate functions more effectively (Pérez-Salinas et al., 2020; Pérez-Salinas et al., 2021; Yu et al., 2022b; Manzano et al., 2023; Jerbi et al., 2023b; Yu et al., 2023). Such characteristics make the data re-uploading QNN a suitable quantum machine model for supervised learning tasks.

Despite these advancements in quantum machine learning, critical challenges remain, particularly in understanding and predicting the performance of models in practical settings. A fundamental criterion for evaluating the performance of any learning algorithm is the generalization gap (Kawaguchi et al., 2022). This gap essentially measures the difference between a model's accuracy on training data and its expected accuracy on unseen data, providing essential theoretical guidance on determining

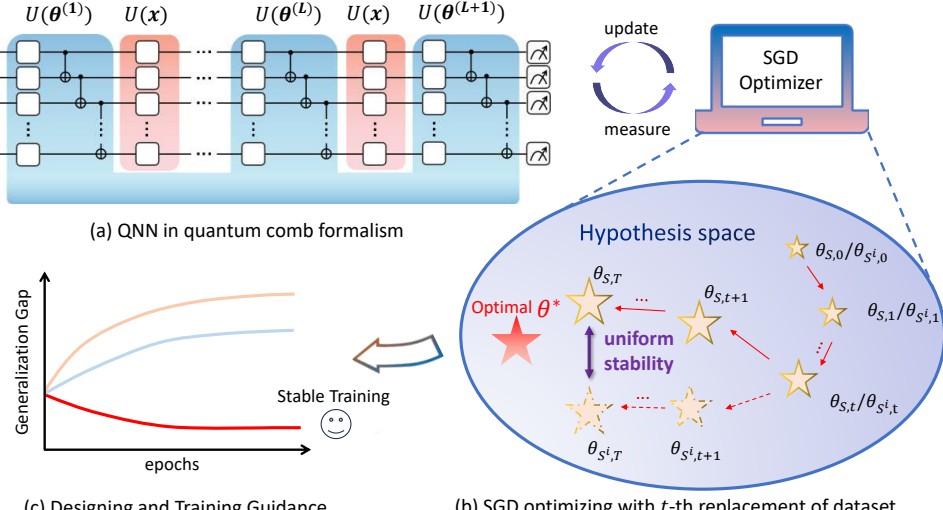

Figure 1: Overview of our work. (a) We investigate the relationship between QNNs and quantum combs, highlighting how this connection informs our understanding of QNN dynamics. (b) We demonstrate that QNNs utilizing the SGD algorithm and trained for $T$ iterations exhibit $\beta_m$-uniform stability. This stability metric quantifies the maximum change in the QNN's output due to alterations in a single training example. (c) Leveraging the uniform stability, we derive a generalization error bound that assesses the QNN's performance on unseen data, thereby better understanding the performance of QNNs in practical applications.

the amount of training data required and informing the design of the model's architecture to ensure robust generalization capabilities.

QNNs typically update trainable parameters using classical optimizers, where classical optimizers play a crucial role in the training process by adjusting the parameters based on quantum measurement results. However, the integration of various classical optimizers with quantum processes introduces additional complexity, particularly in ensuring that these updates contribute positively to minimizing the generalization gap. Hence, providing theoretical guidance on the design and training of QNNs with regard to the strategic use of specific optimizers is of fundamental importance. Among these optimizers, stochastic gradient descent (SGD) algorithm is the most commonly used. Understanding the effects of SGD is vital as it not only enhances our comprehension of this optimizer but also informs the uses of its advanced variants in QNN applications.

To deepen our understanding of QNNs, our work establishes theoretical guarantees on how classical optimizers impact their generalization performance. To summarize, our contributions include:

- We introduce a more general perspective in the study of QNNs by conceptualizing them as a special form of trainable quantum combs (cf. Sec. 3.1). This approach allows us to leverage the rich theoretical framework of quantum combs to analyze the properties and dynamics of QNNs.

- We then investigate the stability of QNNs based on the quantum comb perspective to show that QNNs are uniformly stable (cf. Fig.1). We further establish an upper bound on generalization error for data reuploading model (cf. Sec.3.3). This bound not only informs the control of QNN expressivity power through the number of layers and trainable parameters during the designing of QNNs, but also offers new insights into training QNNs with input data dimensions and classical optimizer-dependent parameters. Importantly, the bound introduces a new training guideline suggesting that the learning rate and the number of trainable gates should be designed to be inversely proportional to enable stable training.

- To substantiate our theoretical claims, we have conducted extensive numerical experiments focused on assessing changes in the expressivity and stability of QNNs. These experiments underscores the importance of stable learning for the practical training of these networks and hence draw a guideline for future QNN developments.

## 1.1 RELATED WORK

Currently, the theory of generalization in QNNs mainly focuses on the complexity measures. Du et al. (2022) derives an upper bound for generalization error with the dependence of the number of trainable quantum gates and the operator norm of the observable by leveraging covering number (Vapnik, 2013) to quantify the expressivity of VQAs. Later, (Caro et al., 2022a) uses quantum channels to derive more general results and (Du et al., 2023) extends it to multi-class classification tasks. Abbas et al. (2021b) uses the effective dimension (Berezniuk et al., 2020; Abbas et al., 2021a) as another complexity measure for the parameter space of QNN. Gyurik et al. (2023) uses Vapnik-Chervonenkis dimension (Vapnik & Chervonenkis, 2015) that investigates the balance of empirical and generalization performance on the dimension of inputs and the Frobenius norm of the observable. Besides, Caro et al. (2021), Bu et al. (2021; 2022) and Qi et al. (2023) derives an upper bound based on Rademacher complexity (Bartlett & Mendelson, 2002), where the bound in (Bu et al., 2021; 2022) is based on the circuit depth and the amount of non-stabilizerness in the circuit for both noise and noiseless models, and Qi et al. (2023) mainly investigates the generalization of variational function regression models using tensor-network encodings. Furthermore, the information-theoretic bound (Banchi et al., 2021) based on Rényi mutual information and the generalization behavior of a specific class of QNN (Chen et al., 2021; Kübler et al., 2021; Du et al., 2021; Wang et al., 2021; Huang et al., 2021a) are also established.

The study of generalization bounds in relation to stability in classical machine learning has laid the substantial groundwork for understanding how small changes in the training set can impact the outputs of learning algorithms (Feldman & Vondrak, 2018; Bousquet et al., 2020; Klochkov & Zhivotovskiy, 2021; Yuan & Li, 2024). From seminal contributions by Bousquet and Elisseeff (Bousquet & Elisseeff, 2002), stability has been demonstrated to yield dimension-independent generalization bounds for both deterministic learning algorithms (Mukherjee et al., 2006; Shalev-Shwartz et al., 2010) and randomized approaches such as stochastic gradient descent (SGD) (Elisseeff et al., 2005; Hardt et al., 2016; Verma & Zhang, 2019). However, despite these significant advances, the analysis of generalization guarantees from the perspective of stability remains largely unexplored in the context of QNNs. Also, existing generalization bounds for QNNs do not account for the impact of the classical optimizer, leading to a significant gap in unified designing and training guidance for effectively implementing powerful QNNs.

## 2 PRELIMINARY

### 2.1 QUANTUM COMPUTING BASICS AND NOTATIONS

**Notations.** We use $\|\cdot\|_p$ to denote the $l_p$-norm for vectors and the Schatten-$p$ norm for matrices. $A^\dagger$ is the conjugate transpose of matrix $A$ and $A^T$ is the transpose of $A$. $\mathrm{tr}[A]$ represent the trace of $A$. The $\mu$-th component of the vector $\boldsymbol{\theta}$ is denoted as $\theta^{(\mu)}$ and the derivative with respect to $\theta^{(\mu)}$ is denoted as $\frac{\partial}{\partial \theta^{(\mu)}}$. We employ $\mathcal{O}$ as the asymptotic notation of upper bounds.

**Quantum state.** In quantum computing, the basic unit of quantum information is a quantum bit or qubit. A single-qubit pure state is described by a unit vector in the Hilbert space $\mathbb{C}^2$, which is commonly written in Dirac notation $|\psi\rangle = \alpha|0\rangle + \beta|1\rangle$, with $|0\rangle = (1,0)^T$, $|1\rangle = (0,1)^T$, $\alpha, \beta \in \mathbb{C}$ subject to $|\alpha|^2 + |\beta|^2 = 1$. The complex conjugate of $|\psi\rangle$ is denoted as $\langle\psi| = |\psi\rangle^\dagger$. The Hilbert space of $N$ qubits is formed by the tensor product "$\otimes$" of $N$ single-qubit spaces with dimension $d = 2^N$. General mixed quantum states are represented by the density matrix, which is a positive semidefinite matrix $\rho \in \mathbb{C}^{d \times d}$ subject to $\mathrm{tr}[\rho] = 1$.

**Quantum gate.** Quantum gates are unitary matrices, which transform quantum states via matrix-vector multiplication. Common single-qubit rotation gates include $R_x(\theta) = e^{-i\theta X/2}$, $R_y(\theta) = e^{-i\theta Y/2}$, $R_z(\theta) = e^{-i\theta Z/2}$, which are in the matrix exponential form of Pauli matrices,

$$X = \begin{pmatrix} 0 & 1 \\ 1 & 0 \end{pmatrix}, \qquad Y = \begin{pmatrix} 0 & -i \\ i & 0 \end{pmatrix}, \qquad Z = \begin{pmatrix} 1 & 0 \\ 0 & -1 \end{pmatrix}. \tag{1}$$

Common two-qubit gates include controlled-X gate $\mathrm{CX} = I \oplus X$ ($\oplus$ is the direct sum), which can generate quantum entanglement among qubits.

## 2.2 Quantum Neural Network

The quantum neural network typically contains three parts, i.e. an $N$-qubit quantum circuit $U(\boldsymbol{\theta}, \boldsymbol{x})$, observables $\mathcal{M} \in \mathbb{C}^{d \times d}$ and a classical optimizer, where $\boldsymbol{x} \in \mathbb{R}^D$ is the encoding of classical data and $\boldsymbol{\theta} \in \mathbb{R}^K$ are trainable parameters. The optimization of the parameterized quantum circuit is based on the feedback from the quantum measurements using the classical optimizer. By assigning a predefined loss function $\ell(\cdot)$, the parameter update rule at iteration $t$ is $\boldsymbol{\theta}_{t+1} = \boldsymbol{\theta}_t - \eta \frac{\partial \ell(f(\boldsymbol{\theta}_t, \boldsymbol{x}, \mathcal{M}), y)}{\partial \boldsymbol{\theta}}$, where $\eta$ is the learning rate, $y$ is the target label, and $f(\cdot)$ is the output of the quantum circuit for the given quantum measurement. The gradient information can be obtained by the parameter shift rule or other methods (Mitarai et al., 2018; Schuld et al., 2019; Stokes et al., 2020) and the design of observables is related to the presense of barren plateaus (Cerezo et al., 2021).

**Basic setup.** In this work, we consider the classification problem, where the training dataset $S := \{z_i = (\boldsymbol{x}_i, y_i)\}_{i=1}^m$ consists of $m = |S|$ samples independently and identically drawn from an unknown probability distribution $\mathbb{D}$. The objective of the machine learning algorithm $\mathcal{A}$ is to utilize $S$ to infer an optimal hypothesis or an optimal classifier $f^*_{\mathcal{A}_S}(\cdot)$ that minimizes the expected risk $R(\mathcal{A}_S) := \mathbb{E}_{(x,y) \sim \mathbb{D}}[\ell(f_{A_S}(\boldsymbol{x}), \boldsymbol{y})]$ (Kawaguchi et al., 2022), considering the inherent randomness in $\mathcal{A}$ and $S$. Given that the probability distribution behind data space $\mathbb{D}$ is generally inaccessible, directly minimizing $\mathcal{R}(\mathcal{A}_S)$ becomes intractable. Consequently, a more practical way involves inferring $f^*(\cdot)$ by minimizing the empirical risk $\hat{\mathcal{R}}_S(\mathcal{A}_S) := \frac{1}{m} \sum_{i=1}^m \ell(f(\boldsymbol{x}_i), y_i)$ on the training dataset $S$. The difference between the empirical risk and the expected risk, known as generalization gap $\mathcal{R}(\mathcal{A}_S) - \hat{\mathcal{R}}_S(\mathcal{A}_S)$, elucidates when and how minimizing $\hat{\mathcal{R}}_S(\mathcal{A}_S)$ effectively approximates the minimization of $\mathcal{R}(\mathcal{A}_S)$.

## 3 Main Result

In this section, we develop our main result of the generalization bound in quantum machine learning models under the discussion of stability. In short, we make connection between quantum comb and data re-uploading QNNs by expressing the output function via a more general form. Then using the characterization of quantum combs in the output function, we derive an upper bound on the generalization gap for data re-uploading QNN that depends on several QNN parameters and hyperparameters that associated with the classical optimizer. We will first introduce the quantum combs and their connection to data re-uploading QNN in Section. 3.1, then we present that QNNs are $\beta_m$-uniform stable in Section. 3.2 and the generalization bound with its implications in Section. 3.3.

### 3.1 Quantum Combs and Data Re-uploading QNNs

A native quantum classifier initially encodes classical data into the quantum state and utilizes a quantum circuit for classification tasks (Mitarai et al., 2018). However, this naive approach faces limitations, even with single qubit classifiers, where a single rotation fails to adequately separate complex data patterns. To overcome these limitations, the data re-uploading QNN is proposed (Pérez-Salinas et al., 2020; Gil Vidal & Theis, 2020), drawing inspiration from classical feed-forward neural networks, in which the classical data is entered and processed in the network several times. The data re-uploading QNN have shown its universality of function approximation (Pérez-Salinas et al., 2020; Pérez-Salinas et al., 2021; Yu et al., 2022b; Manzano et al., 2023) and strong learning performances (Jerbi et al., 2023b; Yu et al., 2023), making it a suitable quantum machine model for supervised learning tasks.

Typically, the model repeatedly encodes classical data into the parameters of quantum gates throughout the quantum circuit. For simplicity, we can assume that the initial input state is denoted as $\rho_{in}$, followed by tunable gates $U(\boldsymbol{\theta})$ that are interspersed with data encoding operations $U(\boldsymbol{x})$. An $L$-layer *data re-uploading QNN* repeats this process $L$ times. We find that the data re-uploading QNN in quantum machine learning is naturally a sequential quantum comb (Chiribella et al., 2008). In the following, we show the output of data re-uploading QNNs $f(\mathcal{C}, x, \mathcal{M})$ with respect to the measurement operator $\mathcal{M}$ in quantum circuits can be well represented by a sequential quantum comb $\mathcal{C}$ with the separation of trainable parts and classical data $\boldsymbol{x}$. Hence, the generalization bound is straightforward to analyze and bounded as shown in Section 3.2.

Using the basic definition of quantum combs and the link product property of Choi–Jamiołkowski isomorphism (Choi, 1975; Jamiołkowski, 1972), we have the following proposition derived from Appendix 8 that characterizes the output of the general quantum comb,

**Proposition 1** *For a general $L$-slot sequential quantum comb $\mathcal{C}$ with classical-data encoding unitary $U(\boldsymbol{x})$ and measurement-channel $\mathcal{M}$, we can represent the output of the quantum comb as*

$$f(\mathcal{C}, \boldsymbol{x}, \mathcal{M}) = \mathrm{tr}\left[\mathcal{C} \cdot \rho_{in}^T \otimes \left(\mathcal{J}_U(\boldsymbol{x})^{\otimes L}\right)^T \otimes \mathcal{M}\right], \tag{2}$$

*where $\mathcal{J}_U(\boldsymbol{x})$ refers to the Choi representation of unitary $U(\boldsymbol{x})$.*

The quantum comb $\mathcal{C}$ can describe a wide range of quantum operations, from simple quantum gates to complex processes that involve multiple steps and interactions and it basically can express any quantum operations that may apply on the QNNs. However, deriving a meaningful generalization bound requires detailed information about the parameters of the QNNs. The general comb formulation, while robust, may not retain all the necessary parameter-specific information needed for this purpose.

We then systematically narrow its scope to investigate the process composition, focusing on a structure with $L$ layers of tunable quantum layers interspersed with classical data encoding channels. In the following, we show the evolution of data re-uploading QNNs can be well represented by a sequential quantum comb with the separation of parameters and re-uploaded data.

**Corollary 2** *For the data re-uploading QNN with depth $L$, we have the output function $f(\boldsymbol{\theta}, \boldsymbol{x}, \mathcal{M})$ in Choi representation as follows:*

$$f(\boldsymbol{\theta}, \boldsymbol{x}, \mathcal{M}) = \mathrm{tr}\left[\bigotimes_{l=1}^{L+1} \mathcal{J}_U(\boldsymbol{\theta}^{(l)}) \cdot \left(\rho_{in}^T \otimes \left(\mathcal{J}_U(\boldsymbol{x})^{\otimes L}\right)^T \otimes \mathcal{M}\right)\right], \tag{3}$$

*where $\mathcal{J}_U(\boldsymbol{\theta}^{(l)})$ denotes the Choi representation of parameterized gates in the $l$-th layer and $\boldsymbol{\theta}^{(l)} \in \boldsymbol{\theta}$ refers to the parameters used in the $l$-th layer.*

This formulation is particularly adept at utilizing quantum dynamics to characterize the output of QNN models. This capability stems from the comb's structured integration of quantum operations, which systematically manipulate and evolve the quantum states based on input data through a series of interconnected dynamics. In the following section, we will demonstrate how this formulation aids in analyzing the stability of QNNs.

## 3.2 SGD STABILITY OF QNNs

Stability in machine learning is determined by analyzing how sensitive the learning algorithm is to modifications in the dataset, such as removing or replacing a data point. For the convenience of this work, we will focus on the scenario where data is replaced and we also note that the concepts of removing and replacing data are essentially interchangeable in the context of stability analysis. This approach allows us to systematically explore how small changes in the dataset influence the performance and reliability of the learning algorithm. Denoting the dataset with replacement on $i$-th data with $z^{'} := (\boldsymbol{x}', y')$ as $S^i$ for $\boldsymbol{x}' \in \mathbb{R}^D$ and $y' \in \mathbb{R}$, where $S^i := \{z_1, \cdots z_{i-1}, z^{'}, z_{i+1}, \cdots z_m\}$, the uniform stability is formally defined as follows:

**Definition 1 (Uniform Stability (Bousquet & Elisseeff, 2002))** *For a randomized learning algorithm $\mathcal{A}_S$, it is said to be $\beta_m$-uniformly stable with respect to a loss function $\ell(\cdot)$, if it satisfies,*

$$\sup_{S,z} |\mathbb{E}_{\mathcal{A}}[\ell(\mathcal{A}_S, z)] - \mathbb{E}_{\mathcal{A}}[\ell(\mathcal{A}_{S^i}, z)]| \leq 2\beta_m. \tag{4}$$

It basically quantifies the maximum change in the output function due to a change in one training example, uniformly over all possible datasets. A randomized algorithm $\mathcal{A}$ is uniformly stable, implying that the models it learns from any two datasets, which differ by only one element, will yield nearly identical predictions across inputs.

In the context of QNNs, the training procedure typically employs stochastic gradient descent (SGD) algorithm (Zhang et al., 2020; Qian et al., 2022b; Sweke et al., 2020). For a given training set $S$, the objective function to be minimized can be expressed as,

$$\min_{\boldsymbol{\theta}} \quad \frac{1}{m}\sum_{i=1}^{m}\ell(f(\boldsymbol{\theta}_S, \boldsymbol{x}_i, \mathcal{M}), y_i). \tag{5}$$

The stochastic gradient update rule for SGD at iteration $t$ is given by

$$\theta_{S,t+1}^{(j)} = \theta_{S,t}^{(j)} - \eta\frac{\partial}{\partial\theta_{S,t}^{(j)}}\ell(f(\boldsymbol{\theta}_{S,t}, \boldsymbol{x}_i, \mathcal{M}), y_i), \quad j = 1, \cdots, K, \tag{6}$$

where $\eta > 0$ is the learning rate, $\boldsymbol{x}_i$ and $y_i$ are the input and output of the randomly selected training example chosen uniformly at each iteration $t$ and $K$ is the number of trainable parameters. The SGD algorithm executes these stochastic gradient updates iteratively, refining the model parameters to minimize the loss over the training set.

We also assume the loss function $\ell$ is Lipschitz continuous and smooth with constants $\mathcal{C}_1$ and $\mathcal{C}_2$ respectively. This assumption is considered to be quite lenient, as the loss functions used in QNNs typically exhibit Lipschitz continuity and are subject to an upper limit defined by the constant. This characteristic is extensively used to investigate the performance capabilities of QNNs (Huang et al., 2021b; Du et al., 2022; McClean et al., 2018; Yu et al., 2023). Drawing on concepts from stability investigations in classical neural networks (Verma & Zhang, 2019; Hardt et al., 2016), we present a proof sketch that demonstrates the uniform stability of the QNN.

**Theorem 3 (Uniform stability of data reuploading QNN)** *Assume the loss function $\ell$ is Lipschitz continuous and smooth. An L-layer data re-uploading QNN trained using the SGD algorithm for $\mathcal{T}$ iterations is $\beta_m$-uniformly stable, where*

$$\beta_m \leq \frac{LD\|\mathcal{M}\|_\infty}{m}\mathcal{O}\left((\eta K\|\mathcal{M}\|_\infty)^{\mathcal{T}}\right). \tag{7}$$

*$K$ denotes the number of trainable parameters in the model, $\mathcal{M}$ is the selected measurement operator, $\eta$ is the learning rate, $m$ refers to the size of the training dataset, and $D$ is the dimension of the data.*

*Proof Sketch.* A sketch version of the proof is as follows, with the details in Appendix 10. In order to prove this theorem, we analyze the output of the QNNs when evaluated on two datasets, $S$ and $S^i$, which differ by a single sample. Given that the loss function is Lipschitz continuous for each example $z_i$, we leverage the linearity of the expectation and the proposition of the difference in the output function (cf. Appendix 10, Lemma S4), we have,

$$|\mathbb{E}_{\text{SGD}}[\ell(\mathcal{A}_S, z) - \ell(\mathcal{A}_{S^i}, z)]| \leq \mathcal{C}_1\mathbb{E}_{\text{SGD}}[|f(\boldsymbol{\theta}_{S,t}, \boldsymbol{x}, \mathcal{M}) - f(\boldsymbol{\theta}_{S^i,t}, \boldsymbol{x}, \mathcal{M})|]$$

$$\leq 2\mathcal{C}_1\|\mathcal{M}\|_\infty\sum_{k=1}^{K}\mathbb{E}_{\text{SGD}}[|\theta_{S,t}^{(k)} - \theta_{S^i,t}^{(k)}|]. \tag{8}$$

Therefore, it is sufficient to analyze how the parameters $\theta_{S,t}$ and $\theta_{S^i,t}$ diverge and bound the change in parameters recursively in the function of iteration $t$.

Based on the training process of SGD, there are two cases to consider the change of parameters. The first case is that SGD selects the example at step $t$ that is identical in $S$ and $S^i$ with probability $(m-1)/m$. Since the parameters $\theta_S$ and $\theta_{S^i}$ may differ and the gradient will also differ, we provide an upper bound for the loss-derivative function,

$$\left|\frac{\partial}{\partial\theta_{S,t}^{(j)}}\ell\left(f(\boldsymbol{\theta}_{S,t}, \boldsymbol{x}, \mathcal{M}), y\right) - \frac{\partial}{\partial\theta_{S^i,t}^{(j)}}\ell\left(f(\boldsymbol{\theta}_{S^i,t}, \boldsymbol{x}, \mathcal{M}), y\right)\right| \leq 2\mathcal{C}_2\|\mathcal{M}\|_\infty\sum_{k=1}^{K}|\theta_{S,t}^{(k)} - \theta_{S^i,t}^{(k)}|. \tag{9}$$

The derivation is based on the parameter change bound (cf. Appendix 10, Lemma S5) and a full derivation is in Lemma S5.

The other case is that SGD selects one example to update in which $S$ and $S^i$ differ and it happens with probability $1/m$. Following the same logic, one can similarly derive the difference loss-derivative function with respect to the different samples as follows,

$$\left| \frac{\partial}{\partial\theta_{S,t}^{(j)}}\ell\left(f(\boldsymbol{\theta}_{S,t},\boldsymbol{x},\mathcal{M}),y\right) - \frac{\partial}{\partial\theta_{S^i,t}^{(j)}}\ell\left(f(\boldsymbol{\theta}_{S^i,t},\boldsymbol{x}',\mathcal{M}),y'\right) \right| \leq 2\mathcal{C}_2\|\mathcal{M}\|_\infty(\sum_{k=1}^{K}|\Delta\theta^{(k)}|+\sum_{j=1}^{LD}|\Delta x^{(j)}|),$$

where D is the data dimension, and we adopt the short notations $|\Delta\theta^{(k)}| = |\theta_{S,t}^{(k)} - \theta_{S^i,t}^{(k)}|, |\Delta x^{(j)}| = x^{(j)} - x'^{(j)}|$, respectively. A full derivation is in Lemma S6. Finally, by taking the probability into consideration and plugging the above results into equation 8, one arrives at equation 7. ∎

Recall that an algorithm is considered stable when the value of $\beta_m$ diminishes in proportion to $1/m$ (Bousquet & Elisseeff, 2002). Accordingly, Theorem 3 explicitly demonstrates that QNNs trained using SGD algorithms exhibit uniform stability. This is characterized by the bounded maximum change in the output function in response to altering a single training example, applicable uniformly across all possible datasets, with the bound scaling as $1/m$. In Section 3.3, we will further explore the implications of the stability with relation to the generalization error.

## 3.3 GENERALIZATION ERROR BOUND AND ITS IMPLICATIONS

The generalization error essentially measures the difference between a model's accuracy on training data and its expected accuracy on new data, serving as a key indicator of the model's performance. Here, we use uniform stability to derive the *generalization bound* of QNNs. Especially, we apply Theorem 3 to obtain the following corollary, with the proof detailed in Appendix 10.

**Corollary 4 (SGD-dependent Generalization Gap)** *Assume the loss function $\ell$ is Lipschitz continuous and smooth. Consider a learning algorithm $\mathcal{A}_S$ that uses the data re-uploading QNN, trained on the dataset $S$ using stochastic gradient descent optimization algorithm over $\mathcal{T}$ iterations. Then, the expected generalization error of $\mathcal{A}_S$ is bounded as follows, holding with probability at least $1 - \delta$ for $\delta \in (0, 1)$,*

$$\mathbb{E}_{SGD}[R(\mathcal{A}_S) - \hat{R}(\mathcal{A}_S)] \leq \frac{LD\|\mathcal{M}\|_\infty}{m}\mathcal{O}\left((\eta K\|\mathcal{M}\|_\infty)^\mathcal{T}\right)$$
$$+ \left(LD\|\mathcal{M}\|_\infty\mathcal{O}\left((\eta K\|\mathcal{M}\|_\infty)^\mathcal{T}\right) + M\right)\sqrt{\frac{\log\frac{1}{\delta}}{2m}}, \tag{10}$$

*where $K$ denotes the number of trainable parameters in the model, $\mathcal{M}$ is the selected measurement operator, $\eta$ is the learning rate, $m$ refers to the size of the training dataset, $D$ is the dimension of data and $M$ is a constant depending on the loss function.*

Corollary 4 directly connects the generalization error with respect to the dimension of datasets, the number of layers, and the number of trainable parameters of data re-uploading QNN. We also consider the depolarizing noise effect on the generalization error, which we remain in the Appendix 12. Corollary 4 also provides additional implications for choosing a suitable learning rate and the number of iterations for training with comparable performance as following.

**Vanishing on the number of samples.** Our generalization bound demonstrates a $\mathcal{O}(\frac{1}{\sqrt{m}})$ scaling with the number of training samples $m$, highlighting that an increase in $m$ directly enhances the generalization performance. It is important to note that for the generalization bound to be meaningful, it must converge to zero as $m$ in the limit $m \rightarrow \infty$ and this convergence is contingent upon $\beta_m$ decaying at a rate faster than $\mathcal{O}(\frac{1}{\sqrt{m}})$. Hence, our generalization bound is a non-trivial bound and its scaling in relation to the number of samples aligns with current literature.

**Trade-off between expressivity and generalization.** On the design of QNNs' architecture, our findings also reveal that the generalization bound demonstrates a linear dependence on both the number of data re-uploading times $L$, and the dimension of the data $D$. This linear relationship indicates that as the number of times data $L$ is re-uploaded increases, or as the dimension of the data $D$ grows, there is a corresponding increase in the bound, suggesting that more complex data

or more frequent re-uploading could potentially lead to larger errors on unseen data. However, the increase in the number of data re-uploading times $L$ implies a higher expressivity of QNNs (Pérez-Salinas et al., 2020; Yu et al., 2022b; Manzano et al., 2023). This directly comes with a trade-off between the expressivity and generalization of data re-uploading QNNs, particularly when dealing with limited training data, which shows the analogy with the bias-variance trade-off in classical neural networks (Geman et al., 1992; Hastie et al., 2009). It is noted that while previous work (Du et al., 2021; Qi et al., 2023) considers the number of parameters as a measure of expressivity, bound which focus on the data re-uploading times $L$ poses a more direct form of expressivity. Below, we demonstrate how the combination of explicit expressivity and optimizer parameters provides additional insights into understanding the performance of QNN models.

**Stable training.** On the training of QNNs, our generalization bound also have an exponential dependence $\mathcal{T}$ on the operator norm of the measurement operator $\|\mathcal{M}\|_\infty$, the learning rate $\eta$ and the number of parameters $K$. It is imperative to carefully choose these parameters such that the term is normalized to be less than 1, i.e. $\eta K\|\mathcal{M}\|_\infty < 1$ to maintain a low generalization error. In practice, the operator norm of measurement practically is bounded by 1 such that $0 \leq \|M\|_\infty \leq 1$ and it is normally chosen as Pauli strings, i.e. $\|X\|_\infty = \|Y\|_\infty = \|Z\|_\infty = 1$. Then, balancing the number of parameters and the learning rate is critical. A larger $K$ can be counterbalanced by a smaller $\eta$ to ensure that each step in the learning process is small enough to prevent instabilities that could arise from complex models. To optimize generalization performance with a fixed number of parameters or learning rate, our bound suggests configuring the learning rate as $\mathcal{O}(1/K)$, where $K$ is the fixed number of parameters, or setting the number of parameters as $\mathcal{O}(1/\eta)$, where $\eta$ is the fixed learning rate. This careful tuning helps to mitigate the risk of overfitting by allowing the model to explore the parameter space more thoroughly and settle into a stable configuration that generalizes well. By incorporating expressivity, our generalization bound unifies the learning and training phases, offering more practical insights for the design and optimization of QNNs.

## 4 Numerical Simulations

Previous sections theoretically characterize the generalization of QNNs via stability using SGD optimizers. In this section, we verify these results by conducting numerical experiments on the perspectives of varies in expressivity and the status of stable training. All evaluations are performed on a desktop with Intel Core i5 CPU (1.4 GHz and 8GB RAM) using python 3.8.

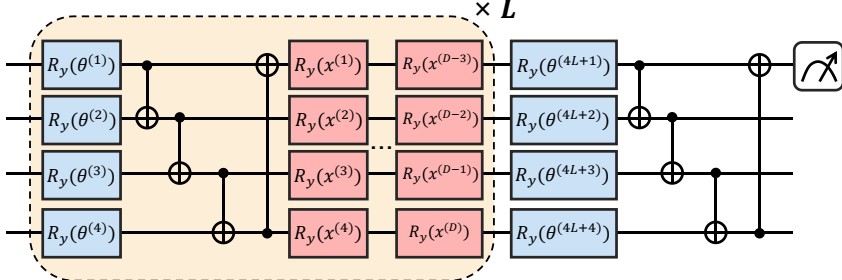

Figure 2: This ansatz template describes a quantum circuit with $N$ qubits and $L$ layers. Each layer is structured into two distinct blocks: a trainable block and an encoding block, represented in blue and red, respectively. The trainable blocks consist of repeated $R_y$ rotation gates and CX gates. In contrast, each encoding block comprises an $n$-tensor product of $R_y(x^{(i)})$ gates, designed to encode an $D$-dimensional classical data vector $\boldsymbol{x} = (x^{(1)}, \cdots, x^{(D)})^T$.

**Simulation Setups.** Following the seminal QML benchmark study (Bowles et al., 2024), we choose to use three public datasets: Breast Cancer (Wolberg, 1995), MNIST (LeCun et al., 2010), and Fashion MNIST (Xiao et al., 2017) to examine the generalization ability via SGD optimizer on data re-uploading QNN. The Breast Cancer dataset has 569 examples described by 30 features. (Fashion) MNIST dataset includes a training set of 60,000 examples and a test set of 10,000 examples. Each example is a 28x28 grayscale image, associated with a label from 10 classes. For simplicity, each

image is reduced to 4x4 for QNNs. We conduct the binary classification tasks on diagnosis B/M, digit 0/1, and class T-shirt/Trouser for the three datasets. The training and testing examples are randomly sampled and the size of training examples are 114, 500, 500, while the size of testing examples are 455, 2000, 2000, respectively. The following table 1 provides a overview of each dataset.

Table 1: Datasets used in experiments

| Dataset | Dimension | Class Number | Training Samples | Testing Samples |
|---|---|---|---|---|
| Breast Cancer | 30 | 2 | 114 | 455 |
| MNIST (4×4 reduced) | 16 | 10 | 500 | 2000 |
| Fashion MNIST (4×4 reduced) | 16 | 10 | 500 | 2000 |

For the architecture of QNNs, we use data re-uploading QNNs with the ansatz in Figure. 2, followed by a Pauli-Z measurement on the first qubit. Adopting the most commonly used ansatz (Kandala et al., 2017; Nakaji & Yamamoto, 2021; Sim et al., 2019), tunable gates are depicted in blue, where the ansatz is considered to be single-qubit rotations with two-qubit CX gates. Also, all trainable parameters are initialized within $[0, 2\pi]$. The classical information $x$ is encoded via angle encoding as depicted in red. We repeated the experiments with varying $L$ settings 5 times and varying $\eta$ settings 10 times to obtain statistical results. The error bars represent the standard deviation.

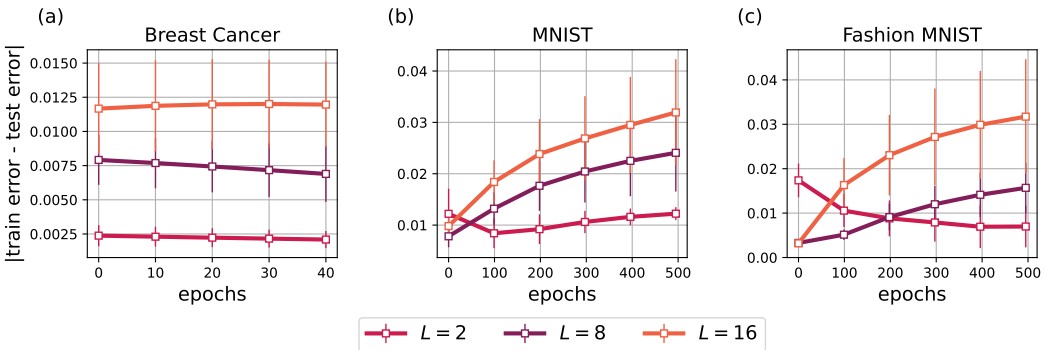

Figure 3: Generalization gap estimation with the varies on the number of data re-uploading change $L \in [2, 8, 16]$ for three datasets with learning rate $\eta = 0.01$, with the error bar representing the statistical uncertainty in experiments. Direct comparisons reveal that the generalization gap is smallest when $L = 2$, whereas, for $L = 8$ or $L = 16$, the gap continues to increase. This observation aligns with our theorem, which suggests that increasing the model's expressivity does not necessarily lead to a convergence in the generalization error.

**Varies on data re-uploading times.** We first study the generalization gap between training and testing loss in binary classification by varying the number of data re-uploading times $L \in [2, 8, 16]$ of the mentioned three datasets with the learning rate $\eta = 0.01$. It is clear to be seen in Figure 3 that as the number of data re-uploading times increases, the generalization gap also increases. Especially with $L = 2$, the generalization bound could achieve relatively better results. This echoes with Theorem 4 that the increase of $K$ and $L$ will increase the generalization gap and it will not be guaranteed to be converged due to the exponential dependence on the iterations.

**Varies on Learning Rate.** We then investigate the generalization gap of stable training by varying the magnitude of learning rate $\eta \in [0.1, 0.05, 0.01]$ of the mentioned three datasets. The number of data re-uploading times is set to be $L = 8$. It is depicted in Figure 4 that with the learning rate $\eta$ increases, the generalization gap is not guaranteed to converge. With the learning rate $\eta = 0.01$, the generalization gap achieves relatively better results as the scaling of the number of parameters closely approximates $\mathcal{O}(1/K)$. However, with higher learning rates such as $\eta = 0.05$ and $\eta = 0.1$, the exponential term is not bounded, implying that the generalization gap may not converge. This observation reinforces the insights from Theorem 4, highlighting the complex relationship between learning rate, parameter scaling, and generalization performance.

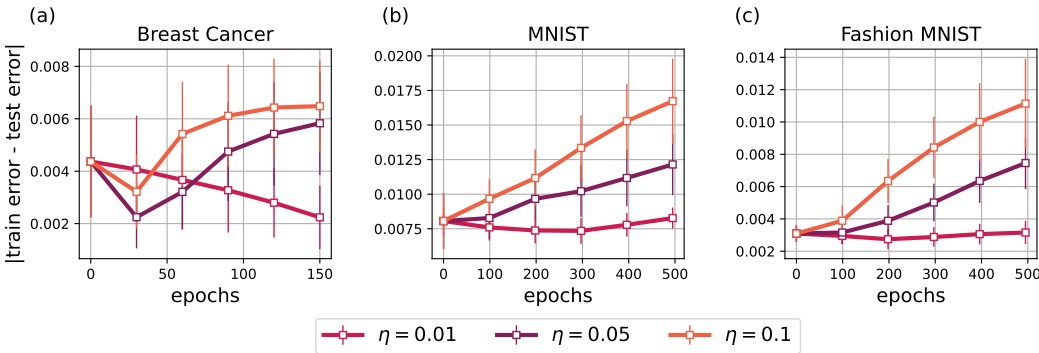

Figure 4: Generalization gap estimation with the varies on the learning rate $\eta \in [0.01, 0.05, 0.1]$ for three datasets, with the error bar representing the statistical uncertainty in experiments. (a), (b), and (c) show the loss of various sampled classical datasets. This figure illustrates that increasing the learning rate does not ensure convergence of the generalization gap. It also demonstrates that the generalization gap tends to converge when the learning rate is chosen to be approximately inversely proportional to the number of parameters.

**Additional Experiments and Remarks.** For completeness, we have also conducted additional experiments on the number of training samples, detailed in Appendix 11 Figure S1. While previous simulations focused on the generalization error gap between training and testing accuracy, we also provide detailed training and testing accuracy results. These results presented in a similar setting to those in Figure 3 and 4 and can be founded in Appendix figure S2 and S4 respectively. We also remark that the generalization bound tends to become too loose as training progresses, particularly in cases where the model's hyperparameters and learning rate are not properly chosen. Observations from several simulations show that the generalization gap converges to a certain value, but the bound fails to capture this phenomenon and instead continues to increase as training progresses.

## 5 CONCLUSION AND FUTURE WORK

In this paper, we have initiated steps towards a deeper theoretical understanding of quantum neural networks by the lens of stability. We have demonstrated that the generalization of data re-uploading QNNs via stability is contingent upon the number of trainable parameters, data re-uploading times, data dimension, and optimizer dependent parameters. While the overall decay of the generalization error in relation to the sample size aligns with results from previous studies, our result provides a distinct perspective on the stable training concept, specifically regarding the impact of learning rate $\eta$, parameter count $K$ and measurement operator $\|M\|_\infty$ on training over $\mathcal{T}$ iterations, given by the exponential term $\mathcal{O}(\eta K \|M\|_\infty)^{\mathcal{T}}$. This offers new insights into training QNNs that the learning rate and the number of trainable gates should be chosen to be inversely proportional to ensure stable training and minimize generalization error.

From the technical perspective, we have utilized the quantum information theoretic methods such as quantum comb architecture and Choi–Jamiołkowski isomorphism to analyze the performance of QNNs. This theoretical framework offers numerous tools for analysing the stability of data re-uploading models. Future research could explore a broader form of the quantum comb and employ semidefinite programming to identify optimal QNNs (Quintino et al., 2019). This approach may offer deeper insights into the fundamental limits that QNNs can achieve in terms of generalization errors. It would also be intriguing to extend our analysis to include other well-known optimizers such as RMSProp and Adam (Ruder, 2016). We believe that our findings deepen the understanding of QNNs' learnability (Anschuetz, 2024) by considering both design and training processes, paving the way for the implementation of powerful QNNs across various machine learning tasks.

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

# 6 APPENDIX

In the table below, we summarize the notations used throughout the paper:

| Symbol | Definition |
| --- | --- |
| $\mathcal{H}_A, \mathcal{H}_{A'}, \cdots$ | Hilbert space of quantum system $A, A', \cdots$ |
| $\mathcal{H}_A^{\otimes n}, \cdots$ | Hilbert space of quantum system $A^n, \cdots$ |
| $\mathrm{Lin}(\mathcal{H}_A)$ | Set of linear operators acting on $\mathcal{H}_A$ |
| id | Identity map on the space $\mathcal{D}(\mathcal{H}_A)$ |
| $I$ | Identity operator on a suitable space |
| $\mathcal{J}_\mathcal{N}$ | Choi representation of the map $\mathcal{N}$ |
| $(\cdot)^T$ | Transpose of an operator |
| $v^{(j)}$ | The $j$-th element of vector $\boldsymbol{v}$ |
| $\boldsymbol{\theta}^{(j)}$ | The $j$-th batch taken from the parameter set $\boldsymbol{\theta}$ |
| $m$ | Numbers of data |
| $D$ | Dimension of data |
| $K$ | Numbers of parameters |
| $L$ | Layers of Quantum Neural Network |
| $\mathcal{T}$ | Iterations |
| $N$ | Number of qubits |
| $\boldsymbol{\theta}_{S,t}$ | Parameters trained from training dataset $S$ after $t$ iterations |
| $\mathcal{M}$ | Measurement operator |
| $\eta$ | Learning rate |

Table 1: Overview of notations.

# 7 AUXILIARY LEMMAS

In this section, we introduce several auxiliary concepts from quantum information theory on quantum circuits or parameterized quantum gates and fundamental definitions from statistical learning theory that are essential for understanding the main proof.

**Lemmas on parameterized quantum gates.** Parameterized quantum circuits (PQCs) are a crucial framework in quantum computing, particularly for variational quantum algorithms. These circuits consist of a sequence of parameterized unitary $U(\boldsymbol{\theta})$ can be adjusted, enabling adaptation to specific tasks or optimization objectives. Common parameterized gates include rotation gates like $R_x(\theta)$, $R_y(\theta)$, and $R_z(\theta)$, which perform rotations on qubits, alongside entangling gates such as the CNOT gate. The parameters are often optimized using classical algorithms to minimize a cost function related to tasks like state preparation or energy estimation. Here we introduce a bound on the distance between two $U(\boldsymbol{\theta})$ with different parameter settings.

**Lemma S1 (Bound in parameters change)** *Suppose a parameterized unitary $U(\boldsymbol{\theta}) := \prod_{k=1}^{K} U_k e^{-i\alpha^{(k)} P_k/2} V_{K+1}$, we have the upper bound on two different parameter sets $\boldsymbol{\alpha}, \boldsymbol{\beta} \in \mathbb{R}^K$*

$$\|U(\boldsymbol{\alpha}) - U(\boldsymbol{\beta})\|_\infty \leq \sum_{k=1}^{K} |\alpha^{(k)} - \beta^{(k)}|, \tag{S1}$$

*where $U_k$ and $V_{K+1}$ are fixed quantum gates, $P_k \in \{X, Y, Z\}$ denotes a single-qubit Pauli gate, and identity gates are omitted.*

**Proof** First, we are going to demonstrate the following inequality:

$$\|U(\boldsymbol{\alpha}) - U(\boldsymbol{\beta})\|_\infty \leq \sum_{k=1}^{K} \left\| e^{-i\alpha^{(k)} P_k/2} - e^{-i\beta^{(k)} P_k/2} \right\|_\infty. \tag{S2}$$

By definition, we have

$$\|U(\boldsymbol{\alpha}) - U(\boldsymbol{\beta})\|_\infty = \left\| \prod_{k=1}^{K} U_k e^{-i\alpha^{(k)} P_k/2} V_{K+1} - \prod_{k=1}^{K} U_k e^{-i\beta^{(k)} P_k/2} V_{K+1} \right\|_\infty. \tag{S3}$$

Without loss of generality, observing the case of $K = 2$, that is,

$$\begin{aligned}
\|U(\boldsymbol{\alpha}) - U(\boldsymbol{\beta})\|_\infty &= \left\| U_1 e^{-i\alpha^{(1)} P_1/2} U_2 e^{-i\alpha^{(2)} P_2/2} V_3 - U_1 e^{-i\beta^{(1)} P_1/2} U_2 e^{-i\beta^{(2)} P_2/2} V_3 \right\|_\infty \\
&\leq \left\| U_1 e^{-i\alpha^{(1)} P_1/2} U_2 e^{-i\alpha^{(2)} P_2/2} V_3 - U_1 e^{-i\beta^{(1)} P_1/2} U_2 e^{-i\alpha^{(2)} P_2/2} V_3 \right\|_\infty \\
&\quad + \left\| U_1 e^{-i\beta^{(1)} P_1/2} U_2 e^{-i\alpha^{(2)} P_2/2} V_3 - U_1 e^{-i\beta^{(1)} P_1/2} U_2 e^{-i\beta^{(2)} P_2/2} V_3 \right\|_\infty \\
&= \left\| e^{-i\alpha^{(1)} P_1/2} - e^{-i\beta^{(1)} P_1/2} \right\|_\infty + \left\| e^{-i\alpha^{(2)} P_2/2} - e^{-i\beta^{(2)} P_2/2} \right\|_\infty,
\end{aligned} \tag{S4}$$

where the inequality follows the triangle inequality, and the last equality refers to the isometric invariance of the spectral norm. Recursively, in the general case of $K \geq 2$, the inequality equation S2 holds.

Additionally, the sub-term $\left\| e^{-i\alpha^{(k)} P_k/2} - e^{-i\beta^{(k)} P_k/2} \right\|_\infty$ in equation S2 can be further bounded

$$\begin{aligned}
\left\| e^{-i\alpha^{(k)} P_k/2} - e^{-i\beta^{(k)} P_k/2} \right\|_\infty &= \left\| I - e^{i(\alpha^{(k)} - \beta^{(k)}) P_k/2} \right\|_\infty \\
&= \max_k \left| 1 - e^{i(\alpha^{(k)} - \beta^{(k)}) \cdot \lambda_k(P_k)/2} \right| \\
&= \max_k \sqrt{2 - 2\cos\left[ (\alpha^{(k)} - \beta^{(k)}) \cdot \lambda_k(P_k)/2 \right]} \tag{S5} \\
&= \sqrt{2 - 2\cos\left[ (\alpha^{(k)} - \beta^{(k)})/2 \right]}, \\
&\leq \left| \alpha^{(k)} - \beta^{(k)} \right|,
\end{aligned}$$

where $\lambda_k(A)$ denotes the $k$-th eigenvalue of a Hermitian operator $A$, the first equation applies the isometric invariance on $\left\| e^{-i\alpha^{(k)} P_k/2} - e^{-i\beta^{(k)} P_k/2} \right\|_\infty = \left\| e^{-i\alpha^{(k)} P_k/2} (I - e^{i(\alpha^{(k)} - \beta^{(k)}) P_k/2}) \right\|_\infty$, and the last inequality holds from the fact that $\cos(\theta) \geq 1 - 2\theta^2$. Finally, plugging this form in the equation S2 yields the result shown in equation S1, which completes the proof. ∎

**Definitions and lemmas from statistical learning theory.** Here, we outline the assumptions underlying our work, provide basic definitions of stability, and show its relationship to the generalization gap.

We will assume the loss function $\ell$ is Lipschitz continuous and smooth with constants $\mathcal{C}_1$ and $\mathcal{C}_2$ respectively. The definition is as follows,

**Definition 2** *A loss function $\ell(\cdot)$ is said to be Lipschitz-continuous and smooth, if it satisfies,*

$$\begin{aligned}
|\ell(f(\cdot), y) - \ell(g(\cdot), y)| &\leq \mathcal{C}_1 |f(\cdot) - g(\cdot)|, \\
|\nabla \ell(f(\cdot), y) - \nabla \ell(g(\cdot), y)| &\leq \mathcal{C}_2 |\nabla f(\cdot) - \nabla g(\cdot)|.
\end{aligned} \tag{S6}$$

This assumption is extensively used to investigate the performance capabilities of QNNs (Huang et al., 2021b; Du et al., 2022; McClean et al., 2018; Yu et al., 2023). Denoting the dataset with replacement on $i$-th data with $z' := (\boldsymbol{x}', y')$ as $S^i$ for $\boldsymbol{x}' \in \mathbb{R}^D$ and $y' \in \mathbb{R}$, where $S^i := \{z_1, \cdots z_{i-1}, z', z_{i+1}, \cdots z_m\}$. The uniform stability is defined as follows,

**Definition 3 (Uniform Stability (Bousquet & Elisseeff, 2002))** *For a randomized learning algorithm $\mathcal{A}_S$, it is said to be $\beta_m$-uniformly stable with respect to a loss function $\ell(\cdot)$, if it satisfies,*

$$\sup_{S, z} |\mathbb{E}_{\mathcal{A}}[\ell(\mathcal{A}_S, z)] - \mathbb{E}_{\mathcal{A}}[\ell(\mathcal{A}_{S^i}, z)]| \leq 2\beta_m. \tag{S7}$$

Uniform stability essentially provides an upper bound on the variation in losses resulting from the alteration of a single data sample. A randomized learning algorithm with uniform stability directly yields the following bound on the generalization error,

**Theorem S2 (Generalization Bound based on Stability (Elisseeff et al., 2005))** *A* $\beta_m$-*uniform stable randomized algorithm with a bounded loss function* $0 \leq \ell(\cdot) \leq M$, *satisfies the following expected generalization bound with probability at least* $1 - \delta$ *with* $\delta \in (0, 1)$ *over the random draw of* $S, z$,

$$E_{\mathcal{A}}[\hat{\mathcal{R}}(\mathcal{A}_S) - \mathcal{R}(\mathcal{A}_S)] \leq 2\beta_m + (4m\beta_m + M)\sqrt{\frac{\log \frac{1}{\delta}}{2m}}. \tag{S8}$$

## 8 BASIC ON QUANTUM COMB

In this section, the method quantum combs (Chiribella et al., 2008) will be introduced. Briefly, let $\mathrm{Lin}(\mathcal{H}_1)$ denote the space of linear operators acting on the Hilbert space $\mathcal{H}_1$, and $\mathrm{Lin}(\mathcal{H}_1, \mathcal{H}_2)$ be the space of linear transforms from $\mathcal{H}_1$ to $\mathcal{H}_2$. It is not trivial to map a linear operator $X \in \mathrm{Lin}(\mathcal{H}_1, \mathcal{H}_2)$ into its vector $\tilde{X} \in \mathcal{H}_1 \otimes \mathcal{H}_2$:

$$\tilde{X} = \sum_{i,j} X_{i,j} |i\rangle \otimes |j\rangle, \tag{S9}$$

where $X_{i,j}$ is the element of $X$ and $|i\rangle, |j\rangle$ are two bases on $\mathcal{H}_1$ and $\mathcal{H}_2$, respectively. Vectorization of linear operator leads to Choi-Jamiołkowski isomorphism of quantum operators. Namely, a CPTP quantum channel $\mathcal{N} \in \mathrm{Lin}(\mathcal{H}_1, \mathcal{H}_2)$ corresponds to its Choi representation $\mathcal{J}_{\mathcal{N}}$ as:

$$\mathcal{J}_{\mathcal{N}} = \mathrm{id}_{\mathcal{H}_1} \otimes \mathcal{N} (\Omega_{\mathcal{H}_1}) = \sum_{i,j} |i\rangle\langle j| \otimes \mathcal{N} (|i\rangle\langle j|), \tag{S10}$$

with $\Omega_{\mathcal{H}_1} = \sum_{i,j} |i\rangle\langle j| \otimes |i\rangle\langle j|$ as the unnormalised maximally entangled state in $\mathcal{H}_1^{\otimes 2}$.

For any two given processes, they can be connected whenever the input system of one matches the output system of the other. In the Choi representations, the composition of two quantum operators $\mathcal{N} \circ \mathcal{M}$ with $\mathcal{M} \in \mathrm{Lin}(\mathcal{H}_0, \mathcal{H}_1)$ and $\mathcal{N} \in \mathrm{Lin}(\mathcal{H}_1, \mathcal{H}_2)$ follows link product, denoted as $\star$:

$$\mathcal{J}_{\mathcal{N} \circ \mathcal{M}} = \mathcal{J}_N \star \mathcal{J}_M = \mathrm{tr}_{\mathcal{H}_1}[(\mathbb{1}_{\mathcal{H}_0} \otimes \mathcal{J}_{\mathcal{N}}) \cdot (\mathcal{J}_{\mathcal{M}}^{T_{\mathcal{H}_1}} \otimes \mathbb{1}_{\mathcal{H}_2})], \tag{S11}$$

with $\mathrm{tr}_{\mathcal{H}_1}$ and $T_{\mathcal{H}_1}$ denotes taking partial trace and transpose on $\mathcal{H}_1$. More properties and detailed discussion about link product and Choi representation are referred to (Chiribella et al., 2009). It is worth mentioning that link product exhibits both associative and commutative properties:

$$\mathcal{J}_1 \star (\mathcal{J}_2 \star \mathcal{J}_3) = (\mathcal{J}_1 \star \mathcal{J}_2) \star \mathcal{J}_3,$$
$$\mathcal{J}_1 \star \mathcal{J}_2 = \mathcal{J}_2 \star \mathcal{J}_1. \tag{S12}$$

A quantum comb is the Choi representation associated with a quantum circuit board and is obtained as the link product of all component operators.

**Lemma S3 (quantum comb (Chiribella et al., 2008))** *Given a matrix* $C \in Lin(\mathcal{P} \otimes \mathcal{I}^n \otimes \mathcal{O}^n \otimes \mathcal{F})$, *it is the Choi representation of a quantum comb* $\mathcal{C}$ *if and only if it satisfies* $C \geq 0$ *and*

$$C^{(0)} = 1, \quad \mathrm{tr}_{\mathcal{I}_i}[C^{(i)}] = C^{(i-1)} \otimes \mathcal{I}_{\mathcal{O}_{(i-1)}}, i = 1, \cdots, n+1, \tag{S13}$$

*where* $C^{(n+1)} := C$, $C^{(i-1)} := \mathrm{tr}_{\mathcal{I}_i \mathcal{O}_{i-1}}[C^{(i)}]/d$, $I_{\mathcal{H}}$ *is the identity operator on* $\mathcal{H}$ *and* $I_{n+1} := \mathcal{F}$, $\mathcal{O} := \mathcal{P}$.

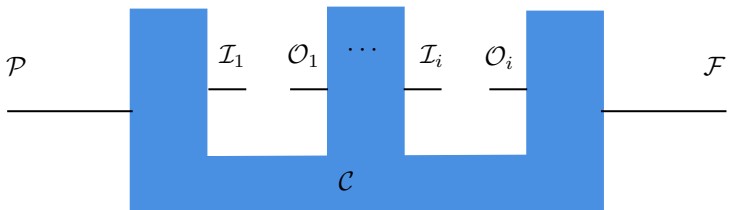

**Proposition 1** *For a general L-slot sequential quantum comb $\mathcal{C}$ with classical-data encoding unitary $U(\boldsymbol{x})$ and measurement-channel $\mathcal{M}$, we can represent the output of the quantum comb as*

$$f(\mathcal{C}, \boldsymbol{x}, \mathcal{M}) = \text{tr}\left[\mathcal{C} \cdot \rho_{in}^T \otimes \left(\mathcal{J}_U(\boldsymbol{x})^{\otimes L}\right)^T \otimes \mathcal{M}\right], \tag{S14}$$

*where $\mathcal{J}_U(\boldsymbol{x})$ refers to the Choi representation of unitary $U(\boldsymbol{x})$.*

**Proof** A general sequential $L$-slot quantum comb $\mathcal{C}$ with $L$ times of input unitary $U(\boldsymbol{x})$ can be represented in link product as

$$\mathcal{C}_U = \mathcal{C} \star \mathcal{J}_U(x)^{\otimes n} = \text{tr}_{\mathcal{I}, \mathcal{O}}\left[C \cdot \left(\mathcal{I}_\mathcal{P} \otimes (\mathcal{J}_U(x)^{\otimes n})^T \otimes \mathcal{I}_\mathcal{F}\right)\right], \tag{S15}$$

where $\text{tr}_{\mathcal{I}, \mathcal{O}}$ refers to partial trace on systems $\{\mathcal{I}_i, \mathcal{O}_i\}_{i=1}^L$. By considering the initial state $\rho_{in}$, we then have the output state $\rho_{out}$ as,

$$\begin{aligned}
\rho_{out} &= \text{tr}_\mathcal{P}[(\rho_{in}^T \otimes \mathcal{I}_\mathcal{F}) \cdot \mathcal{C}_U] \\
&= \text{tr}_\mathcal{P}\left[(\rho_{in}^T \otimes \mathcal{I}_\mathcal{F}) \cdot \text{tr}_{\mathcal{I}, \mathcal{O}}\left[C \cdot \left(\mathcal{I}_\mathcal{P} \otimes (\mathcal{J}_U(x)^{\otimes n})^T \otimes \mathcal{I}_\mathcal{F}\right)\right]\right] \\
&= \text{tr}_{\mathcal{P}, \mathcal{I}, \mathcal{O}}\left[C \cdot \rho_{in}^T \otimes (\mathcal{J}_U(x)^{\otimes n})^T \otimes \mathcal{I}_\mathcal{F}\right].
\end{aligned} \tag{S16}$$

Followed by the further measurement $\mathcal{M}$, we have the output of quantum comb as

$$f(\mathcal{C}, \boldsymbol{x}, \mathcal{M}) = \text{tr}\left[\mathcal{C} \cdot \rho_{in}^T \otimes \left(\mathcal{J}_U(\boldsymbol{x})^{\otimes L}\right)^T \otimes \mathcal{M}\right]. \tag{S17}$$

∎

Overall, quantum combs describe a more general kind of transformation by taking several input operations and output a new operation. It has shown its advantage for applying quantum comb in solving process transformation problems and optimizing the ultimate achievable performance, including transformations of unitary operations such as inversion (Chen et al., 2024; Yoshida et al., 2023), complex conjugation, control-$U$ analysis (Chiribella & Ebler, 2016), as well as learning tasks (Bisio et al., 2010; Sedlák et al., 2019). It can also be used for analyzing more general processes (Zhu et al., 2024) and has also inspired structures like the indefinite causal network (Chiribella et al., 2013; Oreshkov et al., 2012).

## 9 DATA RE-UPLOADING QNN IN CHOI REPRESENTATION

A usual data re-uploading QNN has the following form:

$$\rho_{in} \; \boxed{—1—\; U(\boldsymbol{\theta}^{(1)}) \;—2—\; U(\boldsymbol{x}) \;—3—\; U(\boldsymbol{\theta}^{(2)}) \;—4—\; U(\boldsymbol{x}) \;—5—\cdots—\; U(\boldsymbol{\theta}^{(L)}) \;—\; \measuredangle}$$

$$\tag{S18}$$

For classical data encoding into $U(\boldsymbol{x})$, a universal data re-uploading QNN with circuit depth $L$ can be defined as

$$\begin{aligned}
U(\boldsymbol{\theta}, \boldsymbol{x}) &= U(\boldsymbol{\theta}^{(L+1)})U(\boldsymbol{x})U(\boldsymbol{\theta}^{(L)})\cdots U(\boldsymbol{\theta}^{(2)})U(\boldsymbol{x})U(\boldsymbol{\theta}^{(1)}) \\
&= U(\boldsymbol{\theta}^{(L+1)}) \cdot \prod_{l=1}^L \left(U(\boldsymbol{x})U(\boldsymbol{\theta}^{(l)})\right).
\end{aligned} \tag{S19}$$

where $\boldsymbol{x}$ is the uploaded data and $\boldsymbol{\theta}^{(l)}$ is the $l$-th batch taken from the model parameter set $\boldsymbol{\theta}$.

**Proposition 2** *For the data re-uploading QNN with depth $L$, we have the function $f(\boldsymbol{\theta}, \boldsymbol{x}, \mathcal{M})$ defined as the output of QNN, then in Choi representation followings,*

$$f(\boldsymbol{\theta}, \boldsymbol{x}, \mathcal{M}) = \text{tr}\left[\bigotimes_{l=1}^{L+1} \mathcal{J}_U(\boldsymbol{\theta}_l) \cdot \left(\rho_{in}^T \otimes \left(\mathcal{J}_U(\boldsymbol{x})^{\otimes L}\right)^T \otimes \mathcal{M}\right)\right], \tag{S20}$$

**Proof** As it is mentioned in the Appendix 8, utilizing the link product of Choi representation and its commutative property, we first evaluate $\mathcal{J}_U(\boldsymbol{\theta}^{(2)}) \star \mathcal{J}_U(\boldsymbol{x}) \star \mathcal{J}_U(\boldsymbol{\theta}^{(1)})$. In this proof, it is helpful to label the systems with respect to the indices in equation S18. Notice that

$$\mathcal{J}_U(\boldsymbol{\theta}^{(2)}) \star \mathcal{J}_U(\boldsymbol{\theta}^{(1)}) = \mathcal{J}_U(\boldsymbol{\theta}^{(2)}) \otimes \mathcal{J}_U(\boldsymbol{\theta}^{(1)}). \tag{S21}$$

It can be verified that

$$
\begin{aligned}
\mathcal{J}_U(\boldsymbol{\theta}^{(2)}, \boldsymbol{x}, \boldsymbol{\theta}^{(1)}) &:= \mathcal{J}_U(\boldsymbol{\theta}^{(2)}) \star \mathcal{J}_U(\boldsymbol{x}) \star \mathcal{J}_U(\boldsymbol{\theta}^{(1)}) \\
&= \mathrm{tr}_{2,3}\left[\left[\mathcal{J}_U(\boldsymbol{\theta}^{(1)}) \otimes \mathcal{J}_U(\boldsymbol{\theta}^{(2)})\right] \cdot (I \otimes \mathcal{J}_U^T(\boldsymbol{x}) \otimes I)\right],
\end{aligned}
\tag{S22}
$$

Considering the data encoding with $L$-slot, we recursively derive the Choi representation of the quantum circuit $U(\boldsymbol{\theta}, \boldsymbol{x})$ as $\mathcal{J}(\boldsymbol{\theta}, \boldsymbol{x})$, and

$$\mathcal{J}(\boldsymbol{\theta}, \boldsymbol{x}) = \mathrm{tr}_{2,\ldots,2L+1}\left[\bigotimes_{l=1}^{L+1} \mathcal{J}_U(\boldsymbol{\theta}^{(l)}) \cdot \left(I \otimes \left(\mathcal{J}_U(\boldsymbol{x})^{\otimes L}\right)^T \otimes I\right)\right], \tag{S23}$$

which can be found to match our previous discussion in equation S14. Then, the output function is given as,

$$
\begin{aligned}
f(\boldsymbol{\theta}, \boldsymbol{x}, \mathcal{M}) &= \mathrm{tr}[\mathcal{M} \star \mathcal{J}(\boldsymbol{\theta}, \boldsymbol{x}) \star \rho_{in}] \\
&= \mathrm{tr}\left[\mathcal{J}(\boldsymbol{\theta}, \boldsymbol{x}) \cdot (I_1 \otimes \mathcal{M}) \cdot \left(\rho_{in}^T \otimes I_{2L+2}\right)\right] \\
&= \mathrm{tr}\left[\bigotimes_{l=1}^{L+1} \mathcal{J}_U(\boldsymbol{\theta}^{(l)}) \cdot \left(\rho_{in}^T \otimes \left(\mathcal{J}_U(\boldsymbol{x})^{\otimes L}\right)^T \otimes \mathcal{M}\right)\right].
\end{aligned}
\tag{S24}
$$

This completes the proof. ∎

## 10 PROOF OF MAIN THEORY

We will first present several useful lemmas for deriving the main theorem.

**Lemma S4** *Let $\boldsymbol{\theta}_{S,t}$ and $\boldsymbol{\theta}_{S^i,t}$ represent the parameters learned after $t$ iterations on training sets $S$ and $S^i$, respectively, then the difference in the output function of a data re-uploading QNN is bounded by,*

$$|f(\boldsymbol{\theta}_{S,t}, \boldsymbol{x}, \mathcal{M}) - f(\boldsymbol{\theta}_{S^i,t}, \boldsymbol{x}, \mathcal{M})| \leq 2\|\mathcal{M}\|_\infty \cdot \sum_{j=1}^{K} \left|\theta_{S,t}^{(j)} - \theta_{S^i,t}^{(j)}\right|, \tag{S25}$$

*where $K$ denotes the total number of parameters.*

**Proof** As it is shown in equation S20 from Corollary 2, the difference between the two output functions can be represented in quantum comb formalism as it is shown in equation S20 and further bounded as follows,

$$
\begin{aligned}
&|f(\boldsymbol{\theta}_{S,t}, \boldsymbol{x}, \mathcal{M}) - f(\boldsymbol{\theta}_{S^i,t}, \boldsymbol{x}, \mathcal{M})| \\
&= \left|\mathrm{tr}\left[\left(\bigotimes_{l=1}^{L+1} \mathcal{J}_U(\boldsymbol{\theta}_{S,t}^{(l)}) - \bigotimes_{l=1}^{L+1} \mathcal{J}_U(\boldsymbol{\theta}_{S^i,t}^{(l)})\right) \cdot \left(\rho_{in}^T \otimes \left(\mathcal{J}_U(\boldsymbol{x})^{\otimes L}\right)^T \otimes \mathcal{M}\right)\right]\right| \\
&= \left|\mathrm{tr}\left[\left(\mathcal{J}(\boldsymbol{\theta}_{S,t}, \boldsymbol{x}) - \mathcal{J}(\boldsymbol{\theta}_{S^i,t}, \boldsymbol{x})\right) \cdot (I \otimes \mathcal{M}) \cdot (\rho_{in}^T \otimes I)\right]\right| \\
&\overset{(i)}{=} \left|\mathrm{tr}\left[\left(\mathrm{id} \otimes \mathcal{U}(\boldsymbol{\theta}_{S,t}, \boldsymbol{x})(\Omega) - \mathrm{id} \otimes \mathcal{U}(\boldsymbol{\theta}_{S^i,t}, \boldsymbol{x})(\Omega)\right) \cdot (I \otimes \mathcal{M}) \cdot (\rho_{in}^T \otimes I)\right]\right| \\
&\overset{(ii)}{\leq} \left\|U^\dagger(\boldsymbol{\theta}_{S,t}, \boldsymbol{x})\mathcal{M}U(\boldsymbol{\theta}_{S,t}, \boldsymbol{x}) - U^\dagger(\boldsymbol{\theta}_{S^i,t}, \boldsymbol{x})\mathcal{M}U(\boldsymbol{\theta}_{S^i,t}, \boldsymbol{x})\right\|_\infty \\
&\overset{(iii)}{=} \|U^\dagger(\boldsymbol{\theta}_{S,t}, \boldsymbol{x})\mathcal{M}U(\boldsymbol{\theta}_{S,t}, \boldsymbol{x}) - U^\dagger(\boldsymbol{\theta}_{S,t}, \boldsymbol{x})\mathcal{M}U(\boldsymbol{\theta}_{S^i,t}, \boldsymbol{x})\|_\infty \\
&\quad + \|U^\dagger(\boldsymbol{\theta}_{S,t}, \boldsymbol{x})\mathcal{M}U(\boldsymbol{\theta}_{S^i,t}, \boldsymbol{x}) - U^\dagger(\boldsymbol{\theta}_{S^i,t}, \boldsymbol{x})\mathcal{M}U(\boldsymbol{\theta}_{S^i,t}, \boldsymbol{x})\|_\infty \\
&\overset{(iv)}{=} \left\|\mathcal{M}U(\boldsymbol{\theta}_{S,t}, \boldsymbol{x}) - \mathcal{M}U(\boldsymbol{\theta}_{S^i,t}, \boldsymbol{x})\right\|_\infty + \left\|U^\dagger(\boldsymbol{\theta}_{S,t}, \boldsymbol{x})\mathcal{M} - U^\dagger(\boldsymbol{\theta}_{S^i,t}, \boldsymbol{x})\mathcal{M}\right\|_\infty \\
&\overset{(v)}{\leq} 2\|\mathcal{M}\|_\infty \cdot \left\|U(\boldsymbol{\theta}_{S,t}, \boldsymbol{x}) - U(\boldsymbol{\theta}_{S^i,t}, \boldsymbol{x})\right\|_\infty,
\end{aligned}
\tag{S26}
$$

where $(i)$ recalls the Choi representation of a unitary with $\Omega$ denoting the unormalised maximally entangled state and $\mathcal{U}$ denoting the corresponding quantum channel of the choi representation as equation S10, $(ii)$ uses Hölder's inequality, $(iii), (iv), (v)$ consider triangle inequality, isometric invariance, and submultiplicativity of spectrum norm, respectively. Without loss of generality, assuming all parameters are located on single Pauli rotations, we implement Lemma S1 to complete the proof. ∎

**Lemma S5** *Assume the first order derivative of the loss function $\ell$ be $\mathcal{C}_2$-Lipschitz continuous and smooth. Then, the change in parameter difference of QNNs trained with SGD for $t$ iterations on dataset $S$ and $S^i$ with respect to the same sample can be bounded by,*

$$\left| \frac{\partial}{\partial \theta_{S,t}^{(j)}} \ell\left(f(\boldsymbol{\theta}_{S,t}, \boldsymbol{x}, \mathcal{M}), y\right) - \frac{\partial}{\partial \theta_{S^i,t}^{(j)}} \ell\left(f(\boldsymbol{\theta}_{S^i,t}, \boldsymbol{x}, \mathcal{M}), y\right) \right| \leq 2\mathcal{C}_2 \|\mathcal{M}\|_\infty \sum_{k=1}^{K} |\theta_{S,t}^{(k)} - \theta_{S^i,t}^{(k)}|.$$

**Proof** For convenience, we abbreviate $f(\boldsymbol{\theta}_{S,t}, \boldsymbol{x}, \mathcal{M})$ and $f(\boldsymbol{\theta}_{S^i,t}, \boldsymbol{x}, \mathcal{M})$ as $f_{\boldsymbol{\theta}}(\boldsymbol{x})$ and $f_{\boldsymbol{\theta}^i}(\boldsymbol{x})$, respectively. Denote $\boldsymbol{\theta}_j + \frac{\pi}{2} := (\theta_{S,t}^{(1)}, \cdots, \theta_{S,t}^{(j)} + \pi/2, \cdots, \theta_{S,t}^{(K)})^T$, $\boldsymbol{\theta}_j - \frac{\pi}{2} := (\theta_{S,t}^{(1)}, \cdots, \theta_{S,t}^{(j)} - \pi/2, \cdots, \theta_{S,t}^{(K)})^T$. $\boldsymbol{\theta}_j^i + \frac{\pi}{2}$ and $\boldsymbol{\theta}_j^i - \frac{\pi}{2}$ have similar definitions. Then, we have

$$\left| \frac{\partial}{\partial \theta_{S,t}^{(j)}} \ell\left(f(\boldsymbol{\theta}_{S,t}, \boldsymbol{x}, \mathcal{M}), y\right) - \frac{\partial}{\partial \theta_{S^i,t}^{(j)}} \ell\left(f(\boldsymbol{\theta}_{S^i,t}, \boldsymbol{x}, \mathcal{M}), y\right) \right|$$

$$\overset{(i)}{\leq} \mathcal{C}_2 \left| \frac{\partial}{\partial \theta_{S,t}^{(j)}} f(\boldsymbol{\theta}_{S,t}, \boldsymbol{x}, \mathcal{M}) - \frac{\partial}{\partial \theta_{S^i,t}^{(j)}} f(\boldsymbol{\theta}_{S^i,t}, \boldsymbol{x}, \mathcal{M}) \right|$$

$$\overset{(ii)}{=} \mathcal{C}_2 \left| \frac{1}{2}(f_{\boldsymbol{\theta}_j + \frac{\pi}{2}}(\boldsymbol{x}) - f_{\boldsymbol{\theta}_j - \frac{\pi}{2}}(\boldsymbol{x})) - \frac{1}{2}(f_{\boldsymbol{\theta}_j^i + \frac{\pi}{2}}(\boldsymbol{x}) - f_{\boldsymbol{\theta}_j^i - \frac{\pi}{2}}(\boldsymbol{x})) \right|$$

$$\overset{(iii)}{\leq} \frac{\mathcal{C}_2}{2} \left| f_{\boldsymbol{\theta}_j + \frac{\pi}{2}}(\boldsymbol{x}) - f_{\boldsymbol{\theta}_j^i + \frac{\pi}{2}}(\boldsymbol{x}) \right| + \frac{\mathcal{C}_2}{2} \left| f_{\boldsymbol{\theta}_j - \frac{\pi}{2}}(\boldsymbol{x}) - f_{\boldsymbol{\theta}_j^i - \frac{\pi}{2}}(\boldsymbol{x}) \right|$$

$$\overset{(iv)}{\leq} \mathcal{C}_2 \|\mathcal{M}\|_\infty \left( \|U(\boldsymbol{\theta} + \boldsymbol{\pi/2}) - U(\boldsymbol{\theta}^i + \boldsymbol{\pi/2})\|_\infty + \|U(\boldsymbol{\theta} - \boldsymbol{\pi/2}) - U(\boldsymbol{\theta}^i - \boldsymbol{\pi/2})\|_\infty \right)$$

$$\overset{(v)}{\leq} 2\mathcal{C}_2 \|\mathcal{M}\|_\infty \sum_{k=1}^{K} |\theta_{S,t}^{(j)} - \theta_{S^i,t}^{(k)}|,$$

$$\text{(S27)}$$

where the inequality $(i)$ follows from the condition of Lipschitz continuity and smoothness, equation $(ii)$ is derived from parameter shift rules (Mitarai et al., 2018), and $(iii)$ uses triangle inequality. Taking results from Lemma S1, we have the inequalities $(iv), (v)$ hold, which completes the proof. ∎

**Lemma S6** *Assume the first order derivative of the loss function $\ell$ be $\mathcal{C}_2$-Lipschitz continuous and smooth. Then, the change in parameter difference of L-layers data re-uploading QNNs trained with SGD for $t$ iterations on dataset $S$ and $S^i$ with respect to the different sample can be bounded by,*

$$\left| \frac{\partial}{\partial \theta_{S,t}^{(j)}} \ell\left(f(\boldsymbol{\theta}_{S,t}, \boldsymbol{x}, \mathcal{M}), y\right) - \frac{\partial}{\partial \theta_{S^i,t}^{(j)}} \ell\left(f(\boldsymbol{\theta}_{S^i,t}, \boldsymbol{x}', \mathcal{M}), y'\right) \right| \leq 2\mathcal{C}_2 \|\mathcal{M}\|_\infty \left( \sum_{k=1}^{K} |\Delta\theta_k^i| + \sum_{k=1}^{LD} |\Delta x_k^i| \right),$$

$$\text{(S28)}$$

*where $D$ denotes the dimension of data $\boldsymbol{x}$ and $\boldsymbol{x}'$, $\Delta\theta_k^i := |\theta_{S^i,t}^{(k)} - \theta_{S,t}^{(k)}|$ and $\Delta x_k^i := x'^{(k)} - x^{(k)}$.*

**Proof** We consider the same settings mentioned in Lemma S5. Then, we have

$$
\begin{aligned}
&\left| \frac{\partial}{\partial \theta_{S,t}^{(j)}} \ell \left( f(\boldsymbol{\theta}_{S,t}, \boldsymbol{x}, \mathcal{M}), y \right) - \frac{\partial}{\partial \theta_{S^i,t}^{(j)}} \ell \left( f(\boldsymbol{\theta}_{S^i,t}, \boldsymbol{x}', \mathcal{M}), y' \right) \right| \\
&\overset{(i)}{\leq} \mathcal{C}_2 \left| \frac{\partial}{\partial \theta_j} f_{\boldsymbol{\theta}}(\boldsymbol{x}) - \frac{\partial}{\partial \theta_j^i} f_{\boldsymbol{\theta}^i}(\boldsymbol{x}') \right| \\
&\overset{(ii)}{=} \mathcal{C}_2 \left| \frac{1}{2}(f_{\boldsymbol{\theta}_j + \frac{\pi}{2}}(\boldsymbol{x}) - f_{\boldsymbol{\theta}_j - \frac{\pi}{2}}(\boldsymbol{x})) - \frac{1}{2}(f_{\boldsymbol{\theta}_j^i + \frac{\pi}{2}}(\boldsymbol{x}') - f_{\boldsymbol{\theta}_j^i - \frac{\pi}{2}}(\boldsymbol{x}')) \right| \\
&\leq \frac{\mathcal{C}_2}{2} \left| f_{\boldsymbol{\theta}_j + \frac{\pi}{2}}(\boldsymbol{x}) - f_{\boldsymbol{\theta}_j^i + \frac{\pi}{2}}(\boldsymbol{x}') \right| + \frac{\mathcal{C}_2}{2} \left| f_{\boldsymbol{\theta}_j - \frac{\pi}{2}}(\boldsymbol{x}) - f_{\boldsymbol{\theta}_j^i - \frac{\pi}{2}}(\boldsymbol{x}') \right| \\
&\overset{(iii)}{\leq} \frac{\mathcal{C}_2}{2} \Big( \left| f_{\boldsymbol{\theta}_j + \frac{\pi}{2}}(\boldsymbol{x}) - f_{\boldsymbol{\theta}_j^i + \frac{\pi}{2}}(\boldsymbol{x}) \right| + \left| f_{\boldsymbol{\theta}_j^i + \frac{\pi}{2}}(\boldsymbol{x}) - f_{\boldsymbol{\theta}_j^i + \frac{\pi}{2}}(\boldsymbol{x}') \right| \\
&\qquad + \left| f_{\boldsymbol{\theta}_j - \frac{\pi}{2}}(\boldsymbol{x}) - f_{\boldsymbol{\theta}_j^i - \frac{\pi}{2}}(\boldsymbol{x}) \right| + \left| f_{\boldsymbol{\theta}_j^i - \frac{\pi}{2}}(\boldsymbol{x}) - f_{\boldsymbol{\theta}_j^i - \frac{\pi}{2}}(\boldsymbol{x}') \right| \Big) \\
&\overset{(iv)}{\leq} 2\mathcal{C}_2 \|\mathcal{M}\|_{\infty} \left( \sum_{k=1}^{K} |\Delta \theta_k^i| + \sum_{k=1}^{LD} |\Delta x_k^i| \right),
\end{aligned}
\tag{S29}
$$

where $(i)$ is due to the Lipschitz continuous, $(ii)$ follows from the parameter shift rules, $(iii)$ is taken from triangle inequality and $(iv)$ holds due to Lemma S1. ∎

In the following analysis, we demonstrate that a data re-uploading QNN, when trained using the SGD algorithm, exhibits $\beta_m$ uniform stability. This stability criterion necessitates bounding the difference in the expected value of the learned parameters resulting from a single data perturbation. While this analytical approach aligns with established strategies (Hardt et al., 2016), it is specifically adapted here to the unique context of quantum neural networks.

**Theorem 3** *Assume the loss function $\ell$ is Lipschitz continuous and smooth. A $L$-layer data re-uploading QNN trained using the SGD algorithm for $\mathcal{T}$ iterations is $\beta_m$-uniformly stable, where*

$$
\beta_m \leq \frac{LD\|\mathcal{M}\|_{\infty}}{m} \mathcal{O}\left( (\eta K \|\mathcal{M}\|_{\infty})^{\mathcal{T}} \right).
\tag{S30}
$$

*$K$ denotes the number of trainable parameters in the model, $\mathcal{M}$ is the selected measurement operator, $\eta$ is the learning rate, $m$ refers to the size of the training dataset, and $D$ is the dimension of data.*

**Proof** Using the fact that the loss are Lipschitz continuous, the linearity of expectation and Lemma S4, we have,

$$
\begin{aligned}
|\mathbb{E}_{\text{SGD}}[\ell(\mathcal{A}_S, z) - \ell(\mathcal{A}_{S^i}, z)]| &\leq \mathcal{C}_1 \mathbb{E}_{\text{SGD}}[|f(\boldsymbol{\theta}_{S,t}, \boldsymbol{x}, \mathcal{M}) - f(\boldsymbol{\theta}_{S^i,t}, \boldsymbol{x}, \mathcal{M})|] \\
&\leq 2\mathcal{C}_1 \|\mathcal{M}\|_{\infty} \cdot \sum_{j=1}^{K} \mathbb{E}_{\text{SGD}}[|\theta_{S,t}^{(j)} - \theta_{S^i,t}^{(j)}|].
\end{aligned}
\tag{S31}
$$

Then, we will focus on the term $\sum_{j=1}^{K} \mathbb{E}_{\text{SGD}}[|\theta_{S,t}^{(j)} - \theta_{S^i,t}^{(j)}|]$. Specifically, consider in the training process, SGD optimizer randomly selects a sample that is identical in both training set with probability $(1 - 1/m)$. Then, it will select the sample that is differed in the training set with probability $1/m$. Given an iteration step $t$ and denoting $\Delta \theta_{t+1}^j := \theta_{S,t}^{(j)} - \theta_{S^i,t}^{(j)}$, $f(\boldsymbol{\theta}_{S,t}, \boldsymbol{x}) := f(\boldsymbol{\theta}_{S,t}, \boldsymbol{x}, \mathcal{M})$ for short,

we have the bound on $\mathbb{E}_{SGD}[|\theta_{t+1}^j|]$,

$$
\begin{aligned}
&\mathbb{E}_{SGD}[|\Delta\theta_{t+1}^j|] \\
&\leq (1 - \frac{1}{m})\mathbb{E}_{SGD}[|(\theta_{S,t}^{(j)} - \eta\frac{\partial\ell(f(\boldsymbol{\theta}_{S,t}, \boldsymbol{x}), y)}{\partial\theta_{S,t}^{(j)}}) - (\theta_{S^i,t}^{(j)} - \eta\frac{\partial\ell(f(\boldsymbol{\theta}_{S^i,t}, \boldsymbol{x}), y)}{\partial\theta_{S^i,t}^{(j)}})|] \\
&\quad + \frac{1}{m}\mathbb{E}_{SGD}[|(\theta_{S,t}^{(j)} - \eta\frac{\partial\ell(f(\boldsymbol{\theta}_{S,t}, \boldsymbol{x}'), y')}{\partial\theta_{S,t}^{(j)}}) - (\theta_{S^i,t}^{(j)} - \eta\frac{\partial\ell(f(\boldsymbol{\theta}_{S^i,t}, \boldsymbol{x}''), y'')}{\partial\theta_{S^i,t}^{(j)}})|] \quad\text{(S32)} \\
&= \mathbb{E}_{SGD}[|\Delta\theta_t^j|] + (1 - \frac{1}{m})\eta\mathbb{E}_{SGD}[|\frac{\partial\ell(f(\boldsymbol{\theta}_{S,t}, \boldsymbol{x}), y)}{\partial\theta_{S,t}^{(j)}} - \frac{\partial\ell(f(\boldsymbol{\theta}_{S^i,t}, \boldsymbol{x}), y)}{\partial\theta_{S^i,t}^{(j)}}|] \\
&\quad + \frac{1}{m}\eta\mathbb{E}_{SGD}[|\frac{\partial\ell(f(\boldsymbol{\theta}_{S,t}, \boldsymbol{x}'), y')}{\partial\theta_{S,t}^{(j)}} - \frac{\partial\ell(f(\boldsymbol{\theta}_{S^i,t}, \boldsymbol{x}''), y'')}{\partial\theta_{S^i,t}^{(j)}}|].
\end{aligned}
$$

According to Lemma S5 and Lemma S6 respectively, we have the following two inequalities,

$$
\mathbb{E}_{SGD}[|\frac{\partial\ell(f(\boldsymbol{\theta}_{S,t}, \boldsymbol{x}), y)}{\partial\theta_{S,t}^{(j)}} - \frac{\partial\ell(f(\boldsymbol{\theta}_{S^i,t}, \boldsymbol{x}), y)}{\partial\theta_{S^i,t}^{(j)}}|] \leq 2\mathcal{C}_2\|\mathcal{M}\|_\infty \sum_{k=1}^{K} \mathbb{E}_{SGD}[|\Delta\theta_t^k|] \quad\text{(S33)}
$$

$$
\mathbb{E}_{SGD}[|\frac{\partial\ell(f(\boldsymbol{\theta}_{S,t}, \boldsymbol{x}'), y')}{\partial\theta_{S,t}^{(j)}} - \frac{\partial\ell(f(\boldsymbol{\theta}_{S^i,t}, \boldsymbol{x}''), y'')}{\partial\theta_{S^i,t}^{(j)}}|] \leq 2\mathcal{C}_2\|\mathcal{M}\|_\infty \left(\sum_{k=1}^{K} \mathbb{E}_{SGD}[|\Delta\theta_t^k|] + \sum_{k=1}^{LD} |\Delta x_k^i|\right). \quad\text{(S34)}
$$

Without loss of generality, we set $x^{(k)} \in [0, 2\pi]$, which implies the inequality i.e. $|\Delta x_k^i| \leq 4\pi$. Consequently, we derive the following inequality:

$$
\mathbb{E}_{SGD}[|\Delta\theta_{t+1}^j|] \leq \mathbb{E}_{SGD}[|\Delta\theta_t^j|] + 2\eta\mathcal{C}_2\|\mathcal{M}\|_\infty \sum_{k=1}^{K} \mathbb{E}_{SGD}[|\Delta\theta_t^k|] + \frac{8\pi\eta\mathcal{C}_2\|\mathcal{M}\|_\infty LD}{m}, \quad\text{(S35)}
$$

which also implies that

$$
\sum_{j=1}^{K} \mathbb{E}_{SGD}[|\Delta\theta_{t+1}^j|] \leq (1 + 2\eta\mathcal{C}_2 K\|\mathcal{M}\|_\infty) \sum_{j=1}^{K} \mathbb{E}_{SGD}[|\Delta\theta_t^j|] + \frac{8\pi\eta\mathcal{C}_2 K\|\mathcal{M}\|_\infty LD}{m}. \quad\text{(S36)}
$$

By recursion for each $t$, we have

$$
\sum_{j=1}^{K} \mathbb{E}_{SGD}[|\Delta\theta_T^j|] \leq \frac{8\pi\eta\mathcal{C}_2 K\|\mathcal{M}\|_\infty LD}{m} \sum_{t=1}^{\mathcal{T}} (1 + 2\eta\mathcal{C}_2 K\|\mathcal{M}\|_\infty)^{t-1}. \quad\text{(S37)}
$$

By definition of uniform stability as shown in Definition. 1, we have,

$$
\beta_m \leq \frac{LD\|\mathcal{M}\|_\infty}{m}\mathcal{O}\left((\eta K\|\mathcal{M}\|_\infty)^{\mathcal{T}}\right), \quad\text{(S38)}
$$

which completes the proof. $\blacksquare$

Based on the relationship between uniform stability and the generalization gap, as detailed in Theorem S2, we then establish the following,

**Corollary 4** *Assume the loss function $\ell$ is Lipschitz continuous and smooth. Consider a learning algorithm $\mathcal{A}_S$ that uses the data re-uploading QNN, trained on the dataset $S$ using stochastic gradient descent optimization algorithm over $\mathcal{T}$ iterations. Then, the expected generalization error of $\mathcal{A}_S$ is bounded as follows, holding with probability at least $1 - \delta$ for $\delta \in (0, 1)$,*

$$
\begin{aligned}
\mathbb{E}_{SGD}[R(\mathcal{A}_S) - \hat{R}(\mathcal{A}_S)] \leq &\frac{LD\|\mathcal{M}\|_\infty}{m}\mathcal{O}\left((\eta K\|\mathcal{M}\|_\infty)^{\mathcal{T}}\right) + \\
&\left(LD\|\mathcal{M}\|_\infty\mathcal{O}\left((\eta K\|\mathcal{M}\|_\infty)^{\mathcal{T}}\right) + M\right)\sqrt{\frac{\log\frac{1}{\delta}}{2m}},
\end{aligned} \quad\text{(S39)}
$$

*where $K$ denotes the number of trainable parameters in the model, $\mathcal{M}$ is the selected measurement operator, $\eta$ is the learning rate, $m$ refers to the size of the training dataset, $D$ is the dimension of data and $M$ is a constant depending on the loss function.*

## 11 ADDITIONAL EXPERIMENTS

In this section, we provide additional experimental results on the generalization gap, varying the number of samples. We also present the training and testing accuracies to more clearly illustrate the concepts discussed.

**Varies on number of examples.** We check the convergence of the generalization gap with the increase of training data size $m$ in alignment with Corollary 4. The experiment is implemented by varying the size of train dataset $m \in [100, 500, 1000]$ on MNIST and Fashion MNIST datasets. The size of the testing dataset is set to be 2000. The number of data re-uploading times is set to be $L = 16$, with learning rate $\eta = 0.1$ to achieve better performance on the classification task. It is depicted in Figure S1 that with the examples of training data $m$ increases, the generalization gap is guaranteed to converge.

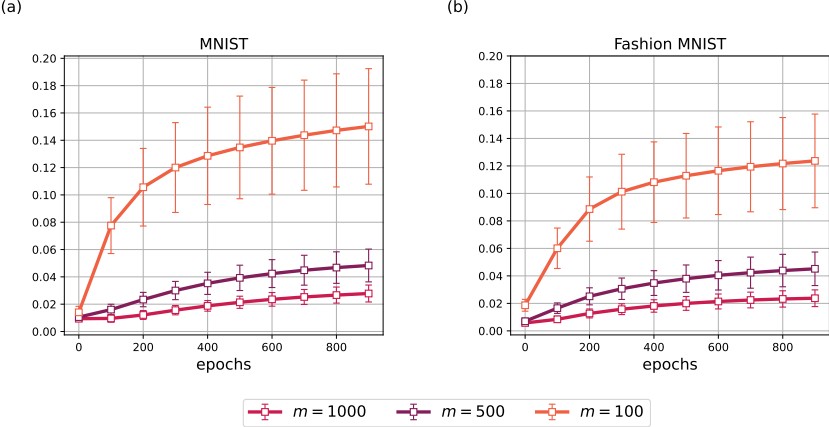

Figure S1: Generalization gap estimation with the varies on a number of examples $m \in [100, 500, 1000]$ for three datasets, with the error bar representing the standard deviation in 5 shots of experiments.

**Training and testing accuracy.** In addition to the experiments presented in the main text, we further demonstrate the training and testing accuracy under settings similar to those depicted in Figures 3 and 4, providing a clearer illustration of performance.

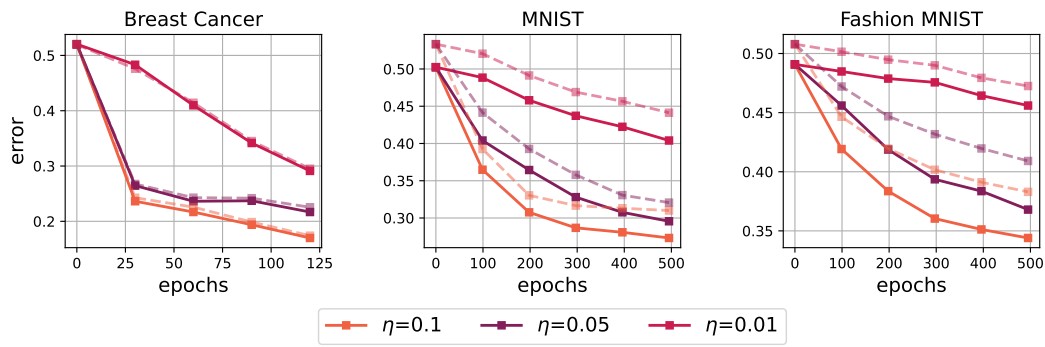

Figure S2: Model performances with varying the learning rate $\eta$ for three datasets, where solid and dashed lines denote training and testing errors, respectively. Errorbars are removed for visibility.

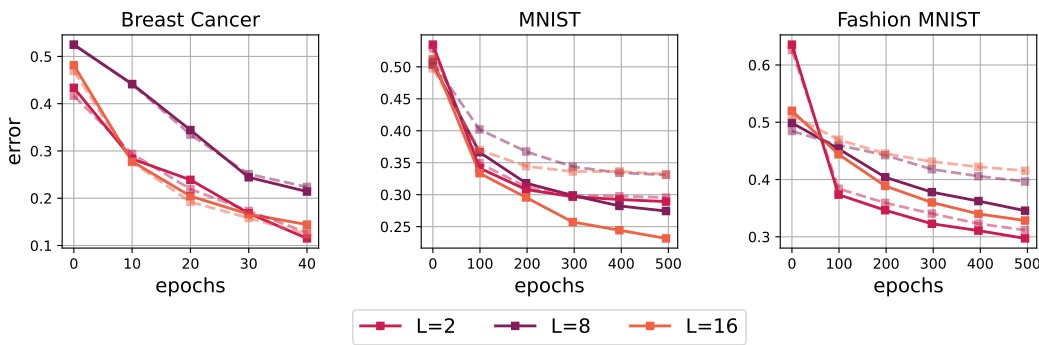

Figure S3: Model performances with varying on the layers $L \in [2, 8, 16]$ for three datasets, where solid and dashed lines denote training and testing errors, respectively.

**Varies of learning rate and data re-uploading times.** In addition to the experiments in the main text, which vary the data re-uploading times from $[2, 8, 16]$ and learning rate from $[0.1, 0.05, 0.01]$, we have conducted additional experiments to further illustrate the transition from stability to instability. In the learning rate experiments, it is observed that as the learning rate increases slightly, the model begins to exhibit unstable behavior. A similar behavior is observed in the experiments varying the number of data re-uploading layers. Additionally, we note that once the model transitions into the unstable phase, the variance increases significantly, which is another characteristic phenomenon associated with instability.

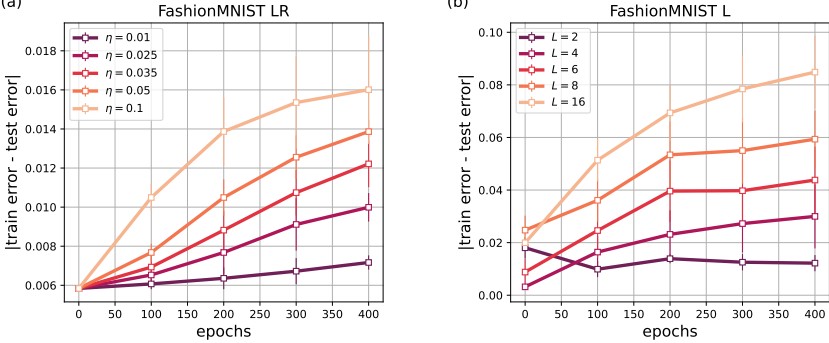

Figure S4: (a) Model performances with varying on learning rate $\eta \in [0.01, 0.025, 0.035, 0.05, 0.1]$. (b) Model performances with varying on the numer of layers $L \in [2, 4, 6, 8, 16]$ for three datasets.

## 12 DEPOLARIZING NOISE

We now consider the scenario where the data re-uploading model is subject to practical noise, specifically the effects of the *local depolarizing channel*. In this setting, a depolarizing channel (Nielsen & Chuang, 2010) $\mathcal{N}_p$ acts after each gate in the data reuploading model. Under this noise model, with probability $1 - p$, the input state remains unchanged, while with probability $p$, the information is lost and the output becomes the maximally mixed state. This noise model effectively simulates practical imperfections in quantum devices and are common in experimental implementations.

The analysis of the robustness of the generalization bound in the presence of standard quantum noise models, such as depolarizing noise, can be easily extended from our previous results. For completeness, we present the following key lemma,

**Lemma S7** *Let $\boldsymbol{\theta}_{S,t}$ and $\boldsymbol{\theta}_{S^i,t}$ represent the parameters learned after $t$ iterations on training sets $S$ and $S^i$, respectively, then the difference in the output function of a data re-uploading QNN that experiencing local depolarizing channel is bounded by,*

$$|f(\boldsymbol{\theta}_{S,t}, \boldsymbol{x}, \mathcal{M}) - f(\boldsymbol{\theta}_{S^i,t}, \boldsymbol{x}, \mathcal{M})| \le 2(1-p)^K \|\mathcal{M}\|_\infty \cdot \sum_{j=1}^{K} \left| \theta_{S,t}^{(j)} - \theta_{S^i,t}^{(j)} \right|, \tag{S40}$$

*where $K$ denotes the total number of parameters.*

**Proof** By combining the lemma from (Du et al., 2021) with the proof technique used Lemma S4, it can be verifed the above lemma holds. ∎

Following a similar procedure as in the proof of the generalization bound, we first examine the effects of depolarizing noise from the perspective of QNN stability,

**Theorem 4** *Assume the loss function $\ell$ is Lipschitz continuous and smooth. A L-layer data re-uploading QNN under local depolarizing noise level $p$ and trained using the SGD algorithm for $\mathcal{T}$ iterations is $\beta_m$-uniformly stable, where*

$$\beta_m \le \frac{(1-p)^{LD} LD \|\mathcal{M}\|_\infty}{m} \mathcal{O}\left( ((1-p)^K \eta K \|\mathcal{M}\|_\infty)^{\mathcal{T}} \right). \tag{S41}$$

*$K$ denotes the number of trainable parameters in the model, $\mathcal{M}$ is the selected measurement operator, $\eta$ is the learning rate, $m$ refers to the size of the training dataset, and $D$ is the dimension of data.*

**Proof** Using the loss are Lipschitz continuous, the linearity of expectation and Lemma S7, we have,

$$|\mathbb{E}_{\text{SGD}}[\ell(\mathcal{A}_S, z) - \ell(\mathcal{A}_{S^i}, z)]| \le \mathcal{C}_1 \mathbb{E}_{\text{SGD}}[|f(\boldsymbol{\theta}_{S,t}, \boldsymbol{x}, \mathcal{M}) - f(\boldsymbol{\theta}_{S^i,t}, \boldsymbol{x}, \mathcal{M})|]$$

$$\le 2(1-p)^K \mathcal{C}_1 \|\mathcal{M}\|_\infty \cdot \sum_{j=1}^{K} \mathbb{E}_{\text{SGD}}[|\theta_{S,t}^{(j)} - \theta_{S^i,t}^{(j)}|]. \tag{S42}$$

Analogous to the proof of Lemma S5 and Lemma S6, we have the following two inequalities which characterize the behavior of the model under local depolarizing noise,

$$\mathbb{E}_{SGD}[|\frac{\partial \ell(f(\boldsymbol{\theta}_{S,t}, \boldsymbol{x}), y)}{\partial \theta_{S,t}^{(j)}} - \frac{\partial \ell(f(\boldsymbol{\theta}_{S^i,t}, \boldsymbol{x}), y)}{\partial \theta_{S^i,t}^{(j)}}|] \le 2(1-p)^K \mathcal{C}_2 \|\mathcal{M}\|_\infty \sum_{k=1}^{K} \mathbb{E}_{SGD}[|\Delta\theta_t^k|] \tag{S43}$$

$$\mathbb{E}_{SGD}[|\frac{\partial \ell(f(\boldsymbol{\theta}_{S,t}, \boldsymbol{x}'), y')}{\partial \theta_{S,t}^{(j)}} - \frac{\partial \ell(f(\boldsymbol{\theta}_{S^i,t}, \boldsymbol{x}''), y'')}{\partial \theta_{S^i,t}^{(j)}}|] \tag{S44}$$

$$\le 2\mathcal{C}_2 \|\mathcal{M}\|_\infty \left( (1-p)^K \sum_{k=1}^{K} \mathbb{E}_{SGD}[|\Delta\theta_t^k|] + (1-p)^{LD} \sum_{k=1}^{LD} |\Delta x_k^i| \right). \tag{S45}$$

By recursion for each $t$ and following the definition of uniform stability as shown in Definition. 1, we have,

$$\beta_m \le \frac{(1-p)^{LD} LD \|\mathcal{M}\|_\infty}{m} \mathcal{O}\left( ((1-p)^K \eta K \|\mathcal{M}\|_\infty)^{\mathcal{T}} \right), \tag{S46}$$

which completes the proof. ∎

**Corollary 5 (SGD-dependent Generalization Gap under Depolarizing Noise)** *Assume the loss function $\ell$ is Lipschitz continuous and smooth. Consider a learning algorithm $\mathcal{A}_S$ that uses the data re-uploading QNN, trained on the dataset $S$ using stochastic gradient descent optimization algorithm over $\mathcal{T}$ iterations under local depolarizing noise level $p$. Then, the expected generalization error of $\mathcal{A}_S$ is bounded as follows, holding with probability at least $1 - \delta$ for $\delta \in (0, 1)$,*

$$\mathbb{E}_{SGD}[R(\mathcal{A}_S) - \hat{R}(\mathcal{A}_S)] \le \frac{(1-p)^{LD} LD \|\mathcal{M}\|_\infty}{m} \mathcal{O}\left( (\eta(1-p)^K K \|\mathcal{M}\|_\infty)^{\mathcal{T}} \right)$$

$$+ ((1-p)^{LD} LD \|\mathcal{M}\|_\infty \mathcal{O}\left( (\eta(1-p)^K K \|\mathcal{M}\|_\infty)^{\mathcal{T}} \right) + M) \sqrt{\frac{\log \frac{1}{\delta}}{2m}}, \tag{S47}$$

where $K$ denotes the number of trainable parameters in the model, $\mathcal{M}$ is the selected measurement operator, $\eta$ is the learning rate, $m$ refers to the size of the training dataset, $D$ is the dimension of data and $M$ is a constant depending on the loss function.

