# OpenReview forum: "Optimizer-Dependent Generalization Bound for Quantum Neural Networks"
_ICLR.cc/2025/Conference — Submitted to ICLR 2025_

### Official Review · Reviewer_D18t · 2024-10-29

**Soundness:** 3
**Presentation:** 3
**Contribution:** 3
**Rating:** 6
**Confidence:** 3

**Summary:**

This paper explores the generalization properties of quantum neural networks (QNNs) with data re-uploading strategy. Concretely, they leverages the quantum combs formalism and data re-uploading structure to analyze the stability of QNNs and then give the upper bound of generalization error on such model. They also provide numerical experiments to support the proposed theoretical claims.

**Strengths:**

1. provide rigorous analysis of generalization performance based on stability of SGD and quantum combs, the upper bound gives some insight to select the hyper-parameters of QML model with data re-uploading.
2. give some basic numerical experiments to support the theoretical claims.

**Weaknesses:**

As the theoretical analysis and numerics only focuses on the specific ansatz structure (with data re-uploading strategy), it is unclear that the similar theoretical findings also true for other QML models.

**Questions:**

I have no further questions.

---

> ### Author Response · Authors · 2024-11-20
> **Response to strengths and weakness**
>
> We sincerely appreciate the reviewer’s kind words about our paper. Below, we provide our responses to the reviewer’s comments.
>
> **Comment 1:** As the theoretical analysis and numerics only focuses on the specific ansatz structure (with data re-uploading strategy), it is unclear that the similar theoretical findings also true for other QML models.
>
> **Reply:** Thank you for taking the time to review our paper and for your comments. We appreciate the opportunity to address your concerns. We would like to first clarify that the theoretical analysis presented in Corollary 4 is not specific to a particular ansatz structure. Instead, it encompasses all ansatz configurations that fall within the data re-uploading framework. The details of the ansatz, such as the number of data re-uploading layers and trainable parameters, are accounted for in the bounds provided in the corollary. In the high level, the model we considered is the data re-uploading model, which also belongs to a broader framework in QML [1] and this model naturally connects to quantum networks with the potential to encompass all possible transformations, including manipulations of quantum states, measurements, and channels [2, 3].
> Furthermore, our results are not restricted to the data re-uploading strategy alone; they can also apply to more commonly used ansatz structures, such as QNNs with a single data encoding layer followed by a tunable layer. For example, by setting the data re-uploading parameter $L=1$, our theoretical findings reduce naturally to these simpler, widely used QNN architectures.
>
>
> Regarding the numerical experiments in Section 4, we acknowledge that they are performed on a specific ansatz structure. The purpose of these experiments is to illustrate and validate the phenomena derived from Corollary 4. We selected a commonly used ansatz to demonstrate the transition of QNNs from stable to unstable regimes by varying parameters such as the number of data re-uploading layers and the learning rate. These findings align well with the theoretical predictions in Corollary 4 and serve as an example of the broader insights provided by our analysis. We hope this explanation clarifies our results and their applicability. We have added the related discussion in the revised version. Thank you again for your constructive feedback.
>
> [1]. Jerbi, Sofiene, et al. "Quantum machine learning beyond kernel methods." Nature Communications 14.1 (2023): 1-8.
>
> [2]. Chiribella, Giulio, Giacomo Mauro D’Ariano, and Paolo Perinotti. "Theoretical framework for quantum networks." Physical Review A—Atomic, Molecular, and Optical Physics 80.2 (2009): 022339.
>
> [3]. Chiribella, Giulio, G. Mauro D’Ariano, and Paolo Perinotti. "Quantum circuit architecture." Physical review letters 101.6 (2008): 060401.

---

### Official Review · Reviewer_2EKn · 2024-10-29

**Soundness:** 3
**Presentation:** 3
**Contribution:** 2
**Rating:** 6
**Confidence:** 2

**Summary:**

The authors derive a generalization bound of the data re-uploading QNN trained using SGD. They first show the data re-uploading QNNs can be represented by a sequential quantum comb, and study the stability of the data re-uploading QNNs via this representation.

**Strengths:**

Understanding behaviors of quantum machine learning models is important. This paper focuses on the combination of a quantum machine learning setting and SGD, and analyzes the generalization gap based on the stability. The topic is interesting and relevent to the community.

**Weaknesses:**

I have several questions and concerns, which are listed below:

- In eq. (10) implies that if we choose a proper hyper parameters, then the generalization gap becomes small as the iteration of SGD proceeds. In many cases, the generalizaiton gap becomes large as the learning process proceeds in an early stage due to the overfitting. On the other hand, the generalization gap may become small after a sufficiently long learning process due to the benign overfitting. Could you comment on the relationship between the proposed generalization bound and this type of practical situations?

- In line 383, the authors instst "the linear dependence of data dimension and data reuploading time not only offers extra insights into understanding the QNN model’s performance but also provides insights into the design of these models". Is it possible to determine an optimal $L$ based on the tradeoff between the expressivity and generalization? Could you explain more about how we can design the models based on the proposed generalization bound?

**Questions:**

Minor comments:
- In eq. (6), the number of parameters are denoted as $K$, but the expranation of this notation is in Theorem 3.

---

> ### Author Response · Authors · 2024-11-20
> **Response to strengths, weakness and questions**
>
> We greatly appreciate the reviewer's kind remarks on the contribution of our results. Below is our point-by-point response to the reviewer's comments.
>
> **Comment 1:** In eq. (10) implies that if we choose a proper hyper parameters, then the generalization gap becomes small as the iteration of SGD proceeds. In many cases, the generalizaiton gap becomes large as the learning process proceeds in an early stage due to the overfitting. On the other hand, the generalization gap may become small after a sufficiently long learning process due to the benign overfitting. Could you comment on the relationship between the proposed generalization bound and this type of practical situations?
>
> **Reply:** Thank you for your insightful question. As noted, in real-world scenarios, the generalization gap often evolves non-monotonically. Our bound in Eq. (10) primarily addresses regimes where the model adequately fits the training data, assuming a relatively small training loss. It does not explicitly capture the underfitting regime, where the training loss remains large, and the model’s expressivity may be insufficient to align with the data distribution. In such cases, standard complexity measures and stability terms used in our bound are less informative, as they rely on the assumption that the training error is relatively low. Thank you for raising this important distinction, which highlights both the limitations and practical relevance of the generalization bound.
>
> **Comment 2:** In line 383, the authors instst "the linear dependence of data dimension and data reuploading time not only offers extra insights into understanding the QNN model’s performance but also provides insights into the design of these models". Is it possible to determine an optimal $L$ based on the tradeoff between the expressivity and generalization? Could you explain more about how we can design the models based on the proposed generalization bound?
>
> **Reply:** Thank you for your thoughtful question and we appreciate the opportunity to clarify this point and its implications for model design. We first want to clarify that the exact evaluation of generalization error is hard, due to the probability distribution behind data space is generally inaccessible. Most of the related results in both classical and quantum domain are focusing on the derivation of the bounds, including our results. Then the optimal $L$ is hard to determine, as the optimal L to these bounds cannot guarantee the optimality for the exact generalization error.
>
>
> However, the generalization bounds we provide are valuable for understanding certain phenomena in QNNs, as they explicitly account for all key components of the model. These bounds can indeed offer practical guidance for model design and hyperparameter tuning in real-world settings. For example, consider training a data reuploading model via SGD on a $D$-dimensional classical datasets with fixed-width and $m$ training datasets. One may ask how many training samples are required to achieve a generalization error within $\epsilon$ using an observable $Z$. By using Corollary 4, one can estimate $ m = \frac{LD}{\epsilon}(\eta K )^T$, providing a rough calculation of the necessary training data. A similar approach can be used to determine how many layers or what learning rate are needed to achieve a specific generalization error when the training dataset is fixed. Thus, these bounds offer valuable estimations for effectively designing and training QNNs.
>
> **Question 1:** In eq. (6), the number of parameters are denoted as $K$, but the expranation of this notation is in Theorem 3.
>
> **Reply:** We thank you for your careful review and valuable feedback. We acknowledge that we omitted a detailed explanation of $K$ in the relevant paragraph. We have now included this explanation in the revised version to address this.

---

> ### Author Response · Authors · 2024-11-26
> **Concluding remark**
>
> We appreciate the reviewer for emphasizing the importance of providing clear guidance on the design of models. Building on these suggestions, we have refined our explanations to address the reviewer's concerns and questions. We hope these explanations meet the reviewer's expectations.

---

> > ### Comment · Reviewer_2EKn · 2024-11-26
> >
> > Thank you for your response. I think the authors properly answered my questions. I understood that some of them are for future work. I will keep my score.

---

### Official Review · Reviewer_cm7s · 2024-11-04

**Soundness:** 3
**Presentation:** 4
**Contribution:** 3
**Rating:** 8
**Confidence:** 3

**Summary:**

This paper attempts to estimate the generalization error bound for Quantum Neural Networks (QNNs) in terms of the number of trainable parameters, data uploading times, dataset dimension and classical optimizer hyperparameters by converting the QNN into a quantum comb structure and by leverage the uniform stability property of the stochastic gradient descent algorithm. The authors then validate these bounds by means of numerical experiments with the Breast Cancer, MNIST and Fashion MNIST datasets by tracking the generalization error being estimated as the absolute difference between the train and test error across epochs for different values of data uploading times and learning rates. The analysis suggests that a larger learning rate should be balanced by a smaller learning rate to ensure that each step in the learning is small enough to prevent instability that could arise in more complex models and that the learning rate should be set $\mathcal{O}(1/K)$, where $K$ is the fixed number of learning parameters or the number of  should be set as $O(1/\eta)$ for a fixed learning rate.

**Strengths:**

The main strength of this paper is that it provides useful guidelines to tune the architecture and learning hyperparameters for QNNs based on the general quantum comb architecture. Each quantum comb is equivalent to a causal QNN. The Choi operator of the quantum comb is the Choi operator of the corresponding causal QNN. Corollary 4 is a useful result that ties all the relevant parameters including number of trainable parameters, data uploading times, dataset dimension and classical optimizer hyperparameters to provide a generalization error bound with SGD-based optimization.

**Weaknesses:**

The main weakness of the paper is the slightly limited experimental evaluation. The choice of datasets are not clearly motivated and only a limited number of values for data uploading times $L$ and the learning rate $\eta$ are tried out, so it is difficult to assess from the plots where we move from stable regime to the unstable regime (see the plot for Fashion MNIST in Figure 1 for an example). Also, the error bars are very large for experiments that vary the learning rate.

**Questions:**

1. Why did you choose the datasets you chose for the experimental evaluation? It would be good to have a table that shows the datasets chosen with the number of training examples.
2. Have you considered the complexity of the learning task (eg. lack of easy separability between the classes) or the number of features in your data as affecting your generalization bounds? How does it factor into your bound provided in Corollary 4 or is it independent of that?
3. For the Fashion MNIST dataset experiment, you may wish to try more values of $L$ for Figure 1 and $\eta$ for Figure 2 because there is rather ubrupt transition from stability to unstability from $L = 2$ to $L = 8$ and from $\eta = 0.01$ to $\eta = 0.05$.
4. The error bounds are rather large for the experiments that vary the learning rate (Figure 2), making the conclusions you are drawing somewhat untenable, you may wish to run more trials to reduce the size of your error bars. I understand you reran 5 times but that maybe too little for some of the experiments. Why did you not do this?

---

> ### Author Response · Authors · 2024-11-20
> **Response to strengths, weakness and questions**
>
> We thanks the reviewer for their kind remarks on the results of our paper. We also appreciate the comments on the numerics part, which could further strength our paper. Below, we will address the questions of the reveiwer, which also address the weakness of the paper that specified by the reviewer.
>
> **Question 1:** Why did you choose the datasets you chose for the experimental evaluation? It would be good to have a table that shows the datasets chosen with the number of training examples.
>
> **Reply:** Thank you for your insightful comment regarding the datasets and the table displaying these. The datasets that we chosen is following the seminal work in ref.[1] and there are many common QML works also focus on this dataset (i.e. ref.[2]). Our motivation for focusing on the dataset is to use it as a well-established benchmark to illustrate the overall phenomenon described in Corollary 4, leveraging its simplicity and widespread use to clearly demonstrate the theoretical insights in a practical and recognizable context. We also agree that including such a table can significantly enhance the reader’s understanding of our experimental setup, thus the table is augmented into the revised manuscript. The table provides the provides a clear overview of the dimensions, class numbers, and the training/testing splits for each dataset.
>
> [1]. Bowles, Joseph, Shahnawaz Ahmed, and Maria Schuld. "Better than classical? the subtle art of benchmarking quantum machine learning models." arXiv preprint arXiv:2403.07059 (2024).
>
> [2]. Qian, Yang, et al. "The dilemma of quantum neural networks." IEEE Transactions on Neural Networks and Learning Systems (2022).
>
> **Question 2:** Have you considered the complexity of the learning task (eg. lack of easy separability between the classes) or the number of features in your data as affecting your generalization bounds? How does it factor into your bound provided in Corollary 4 or is it independent of that?
>
> **Reply:** Thank you for your question. We would appreciate clarification on what specific measure of complexity you are referring to in the context of the learning task. If you are referring to the number of features, we can confirm that this is indeed accounted for in our generalization bounds. As shown in Corollary 4, $D$ represents the dimension of the data, which corresponds to the number of features. Our bound indicates a linear dependence on the number of features, implying that as the number of features increases, it becomes more challenging for the model to achieve a smaller generalization error. This observation also suggests that a larger number of features generally requires more training data to effectively reduce the error.
>
> **Question 3:** For the Fashion MNIST dataset experiment, you may wish to try more values of $L$ for Figure 1 and $\eta$ for Figure 2 because there is rather ubrupt transition from stability to unstability from $L=2$ to $L=8$ and from $\eta=0.01$ to $\eta=0.05$.
>
> **Reply:** Thanks for your comments and we have added more values of $L$ and $\eta$ to better illustrate the transition from stability to unstability in the revised version Appendix section 11. It is observed that as the number of layers or the learning rate increases, the model tends to become slightly unstable. Additionally, we note that with this instability, the variance associated with each unstable training case also increases, which may serve as an indicator of the transition to instability. Thank you for pointing that out and it helps strengthen our paper.
>
> **Question 4:** The error bounds are rather large for the experiments that vary the learning rate (Figure 2), making the conclusions you are drawing somewhat untenable, you may wish to run more trials to reduce the size of your error bars. I understand you reran 5 times but that maybe too little for some of the experiments. Why did you not do this?
>
> **Reply:** Thank you for your valuable feedback. In the current version, we conducted 5 trials, which we believed were sufficient to capture the overall phenomenon and demonstrate the trends predicted by our theoretical analysis. We acknowledge that increasing the number of trials would reduce the variability and make the results more robust. We also observed that unstable training dynamics are typically associated with increased variance, which can serve as an indicator of models with improperly chosen hyperparameters. In response to your suggestion, we have conducted additional experiments to validate this phenomenon and included the updated results in Figure 2 of the revised version.

---

> ### Author Response · Authors · 2024-11-26
> **Concluding remark**
>
> We sincerely thank the reviewer once again for their valuable feedback on our work. Guided by the insightful and well-targeted comments, we have incorporated additional experiments to incorporate the suggestions. We believe these revisions have significantly enhanced the numerical aspects of our paper, and we hope the reviewer finds our updated manuscript to be a substantial improvement over the original submission.

---

> ### Comment · Reviewer_cm7s · 2024-11-28
> **Thank you**
>
> Thank you for your detailed responses and for adding the additional experiments! I will raise my score from 6 to 8. I was referring to separability of the data and number of features. Your bound takes into account number of features, so that is good! It would be good to quantify the complexity of the learning task in terms of separability of the data. It should be noted that the generalization bound is not tight due to the exponent.

---

### Official Review · Reviewer_x6gC · 2024-11-04

**Soundness:** 3
**Presentation:** 3
**Contribution:** 2
**Rating:** 5
**Confidence:** 4

**Summary:**

This paper investigates the generalization properties of quantum neural networks (QNNs) through the lens of learning algorithm stability. The authors establish a connection between quantum combs—a framework for organizing variable subcircuits—and data re-uploading, a technique for iteratively encoding data within quantum circuits. They derive an upper bound on the generalization error, relating it to the number of re-uploading layers, dataset size, parameter count, learning rate, and optimization steps. Experimental results are provided to support the theoretical findings.

**Strengths:**

1. This study initiates the attempt to establish the optimization-dependent generalization error bound of quantum neural networks by analyzing the stability of optimizing quantum neural networks (QNNs) with SGD.
2. The derived generalization bound offers practical insights for selecting hyperparameters, such as recommending a smaller learning rate when using a large number of parameters, which could be beneficial for optimizing QNNs.
3. Numerical experiments are conducted, providing empirical support for the theoretical results and suggesting that the bounds are valid under various settings.

**Weaknesses:**

1. While this paper presents a useful framework for optimization-dependent generalization error bounds, its novelty is somewhat limited in both theoretical derivation and implications. Specifically, two of the three main implications have already been observed in Ref. [1], and the third can be derived from Ref. [2], which discusses model stability under different convexity conditions—a result that may be applied to quantum settings as well.
2. Although it is interesting to establish the connection between the quantum comb and the QNNs with data-reuploading structure, the discussion of this connection seems to be separated from the derivation of the generalization error bound. I mean by this the derivation of the generalization error bound does not employ any necessary properties of the quantum comb. This raises questions about the added value of quantum combs in the context of this study.

[1]. Caro, Matthias C., et al. "Encoding-dependent generalization bounds for parametrized quantum circuits." Quantum 5 (2021): 582.
[2]. Hardt, Moritz, Ben Recht, and Yoram Singer. "Train faster, generalize better: Stability of stochastic gradient descent." International conference on machine learning. PMLR, 2016.

**Questions:**

Could you clarify the role of quantum combs in the derivation of the generalization bounds, specifically whether they are essential to the analysis or if the same results could be obtained without them?

---

> ### Author Response · Authors · 2024-11-20
> **Response to strengths, weakness and questions**
>
> Thank you for your kind words on the optimizer-dependent generalization error bounds. We now clarify the novelty of our work.
>
> **Comment 1:** While this paper presents a useful framework for optimization-dependent generalization error bounds, its novelty is somewhat limited in both theoretical derivation and implications. Specifically, two of the three main implications have already been observed in Ref. [1], and the third can be derived from Ref. [2], which discusses model stability under different convexity conditions—a result that may be applied to quantum settings as well.
>
> **Reply:**  (i) On the technical side, we would like to highlight that previous approaches to analyzing the generalization of QNNs, such as those based on Rademacher complexity, provide bounds that apply uniformly across the entire space of possible functions. However, these bounds may not fully capture the behavior of specific learning algorithms utilized in practice. In contrast, stability offers a different perspective by focusing on the space of particular learning algorithms.
> While there are several works investigating stability in the context of classical neural networks, its application to QNNs remains unclear, particularly in terms of identifying which properties influence their learning dynamics for specific algorithms of interest. The extension to QNNs is not straightforward. The unique characteristics of quantum models, including their parametrization, data encoding, and the influence of quantum observables, introduce challenges that are not directly addressed in classical settings.
>
>
> (ii) On the results side, our work provides a unified description of the three main components of QNNs, i.e. the ansatz, the observable, and the classical optimizer. To the best of our knowledge, this integration has not been explored in the QML literature. While certain implications of our work may resemble findings from prior studies, the interplay between generalization error, expressivity, and hyperparameter choices during training has not been connected in previous research. Our framework offers a new characterization that bridges the design and training of QNNs within a more practical and concrete setting, emphasizing the interdependence of these components. We believe that our optimizer-dependent generalization bound aligns more closely with the practical needs of QNN design and training, thereby appealing to a broader audience in the QML community. We have revised the point more clearly in the revised version.
>
> **Comment 2 and Question 1:** Although it is interesting to establish the connection between the quantum comb and the QNNs with data-reuploading structure, the discussion of this connection seems to be separated from the derivation of the generalization error bound. I mean by this the derivation of the generalization error bound does not employ any necessary properties of the quantum comb. This raises questions about the added value of quantum combs in the context of this study. Could you clarify the role of quantum combs in the derivation of the generalization bounds, specifically whether they are essential to the analysis or if the same results could be obtained without them?
>
> **Reply:** Thank you for your thoughtful comment and for raising this important point regarding the role of quantum combs in our study. We believe that the quantum comb framework provides a valuable perspective for investigating quantum neural networks by establishing a connection through the Choi isomorphism and it enables calculations to be systematically articulated using link products. Specifically, it provides a structured way to address the unique interplay between the data encoding and trainable components of QNNs. By naturally separating these two aspects, the application of this framework could help us analyse of stability and the derivation of generalization bounds. Moreover, the quantum comb formalism could potentially provide additional interpretative value by framing QNNs within a broader theoretical context, which could be useful for extending the analysis to other related architectures in future work. We hope this explanation clarifies the rationale behind using quantum combs in our study and highlights their added value.

---

### Official Review · Reviewer_893S · 2024-11-05

**Soundness:** 3
**Presentation:** 2
**Contribution:** 2
**Rating:** 5
**Confidence:** 5

**Summary:**

In this paper, the authors provide generalization bounds using the _stability_ approach for a family of quantum neural networks (QNNs) using data-reuploading strategy and stochastic gradient descent (SGD) as an optimizer by conceptualizing QNNs as a special form of quantum combs. The generalization bound number of trainable parameters, number of times data is reuploaded, dataset dimension, and classical optimizer-dependent parameters, providing guidelines for designing and training QNNs. Numerical experiments to support theoretical claims have been conducted on three classical datasets; e.g. MNIST, Breast Cancer, and FMNIST.

**Strengths:**

This work makes an interesting connection of QNNs to quantum combs to leverage the rich theoretical framework of the latter to analyze the dynamics of QNNs, although a brief connection of quantum combs to QNNs using data re-uploading strategy was mention in [1].

Provides theoretical guidelines on designing and training near-term quantum machine learning (QML) models--a fundamental question in QML.

[1] Mo, Yin, et al., "Parameterized quantum comb and simpler circuits for reversing unknown qubit-unitary operations", arXiv:2403.03761

**Weaknesses:**

- Limited novelty in theoretical analysis: In my opinion, the paper heavily borrows proof strategies and techniques from [2] which studies SGD for smooth, Lipschitz and convex problems for deep neural networks. The only novelty is the connection of data reuploading QNNs to quantum combs and then using well-established lemmas from the rich theory of quantum combs.

- The generalization bounds based on stability especially that of [2] becomes too loose and vacuous as training progresses, even in the practical regime of the training time where deep neural networks are known to generalize well [3]. I believe that as the generalization bound essentially uses the same proof techniques and the technical setting is the same, your bound might also undergo the similar phenomenon. This is a limitation which should be highlighted in the work.

- The work also lacks analysis on the robustness of the generalization bound in presence of standard quantum noise models such as depolarizing noise, state preparation and measurement error (SPAM), thermal relaxation, readout error etc. It would be beneficial if the authors can provide analysis of their generalization bounds for one of the above noise models, e.g. for depolarization noise.

**Minor Issues**:

- Line 161: the definition of CX gate is incorrect.

- Line 164: A more of a pedantic take, apologies if the authors do not agree with me on this: as QNN is a variational quantum algorithm (VQA), I think they comprise of 3 things: ansatz, observable and the classical optimizer. I say this because the design of observables also is related to the presense of barren plateaus. For instance, [4] proves that when you have local observables there is a possibility of absence of barren plateaus (BPs) while for global observables, they prove the presence of BPs.

- In Fig. 2 caption and in the surrounding text, the authors mention the use angle encoding as a classical data encoding scheme, however, this is not correctly depicted in Fig. 2 as there are two RY gates! Please correct this to avoid any confusion.

- Please add reproducibility statement according to the author guidelines of ICLR.


[2] Hardt, Moritz et al., "Train faster, generalize better: Stability of stochastic
gradient descent", ICML (2016)

[3] Kawaguchi et al., "Generalization in Deep Learning", CUP (2022)

[4] Cerezo, Marco et al., "Cost Function Dependent Barren Plateaus in Shallow Parametrized Quantum Circuits", Nat. Comms. (2021)

**Questions:**

- Can your proof techniques be used for proving generalization bounds for data re-uploading QNNs where input-scaling parameters are also trainable? That is, the output function of a data re-uploading QNN is $f(\mathbf{\theta}, \mathbf{\lambda}, \mathbf{x}, \mathcal{M})$, where $\mathbf{\lambda}$ are input-scaling parameters used when data is being re-uploaded and are trainable, e.g. data encoding operation is $U(\mathbf{x}, \mathbf{\lambda})$ instead of $U(\mathbf{x})$. In your current analysis, $\mathbf{\lambda} = \mathbf{1}$ and is not trainable. I ask this because the function $f(\mathbf{\theta}, \mathbf{\lambda}, \mathbf{x}, \mathcal{M})$ is more general and recovers your defintion of $f(\mathbf{\theta}, \mathbf{x}, \mathcal{M})$ and can in principle do a bit more than your data reuploading model.

- Does your definition of stability hold also for memorization algorithm[5]--a learning algorithm which always outputs the function $f(\mathbf{x}_i, \mathbf{\theta}_S, \mathcal{M}) = -1$, except that it alters $f$ in a small number of locations to fit the training set?

- Can you provide any numerical evidence on the how tight is your analysis of generalization bound?

- What properties (or hyperparameters of QNNs) and operations (regularization, weight decay, gradient clipping etc.) induce stability?

- DNNs with random initialization tend to behave like random Gaussian processes (in limit of many neurons)[6] which allows for their characterization of their trainability and generalization. However, this is not the case for randomly initialized QNNs as they instead generally form Wishart processes[7] which give rise to barren plateau phenomenon and exponentially many local poor minima. That being said, does your generalization bound result help to mitigate these issues and if yes, how?

- Why is there a significant difference between the generalization error reported in Fig. 3 with settings [L=8 (purple or violet) with $\eta=0.01$] and Fig. 4 with settings [L=8 with $\eta=0.01$ (red)]. The generalization error should be more or less similar which is not the case if I look at the plots. Can you please explain this phenomenon?

- In the numerical experiments to support your theoretical claims, why do you use parametrized quantum circuits (PQCs) that are not universal and classically simulable? This completely defeats the purpose of using QML, no? Because at the end, we want a quantum model (specified by a PQC) which is expressive (for the domain), trainable, noise robust and can not be classically simulable--holy grail of QML.



[5] Chatterjee, Satrajit, "Learning and Memorization", ICML (2018)

[6] Lee et al., "Deep neural networks as Gaussian processes", ICLR (2018)

[7] Anschuetz, Eric, "A Unified Theory of Quantum Neural Network Loss Landscapes", arXiv:2408.11901

---

> ### Author Response · Authors · 2024-11-20
> **Response to strengths and weakness.**
>
> We sincerely thank the reviewer for their thorough review and the insightful question they posed.
>
> **Comment 1:** Limited novelty in theoretical analysis.
>
> **Reply:** Thank you for your criticism regarding the novelty of our theoretical analysis. As indicated in our title, our primary goal is to leverage stability as a framework to investigate the effects of classical optimizers on the generalization error in QNNs. This is a topic that, to the best of our knowledge, has not been addressed in QML. While it is true that some proof techniques are borrowed from CNNs, adapting these methods to the quantum setting presents unique challenges, specifically its connection to QNN models. We believe the generalization bounds and the accompanying analysis offer a valuable contribution to the field of QML by addressing previously unexplored aspects of QNN training and optimization.
>
> **Comment 2:** The generalization bounds based on stability especially that of [2] becomes too loose and vacuous as training progresses, even in the practical regime of the training time where deep neural networks are known to generalize well [3]. I believe that as the generalization bound essentially uses the same proof techniques and the technical setting is the same, your bound might also undergo the similar phenomenon. This is a limitation which should be highlighted in the work.
>
> **Reply:** Thank you for raising this important question. We acknowledge that in certain cases, the generalization bound can become too loose. This phenomenon is also reflected in some of our numerical experiments, where improper hyperparameter settings or an excessive number of data re-uploading layers result in unstable training. In these instances, the generalization gap increases and appears to converge to a specific value, which aligns with the bound becoming loose due to the exponential term in our analysis.
>
> However, when proper hyperparameters and an appropriate number of data re-uploading layers are used, the generalization gap decreases as training progresses. In these cases, the generalization bounds and numerical simulations capture the phenomenon effectively, demonstrating that the bounds may remain reasonably tight under well-chosen training conditions. We have highlighted this limitation and the associated phenomenon in the revised version to address your concern.
>
> **Comment 3:** The work also lacks analysis on the robustness of the generalization bound in presence of standard quantum noise models such as depolarizing noise, state preparation and measurement error (SPAM), thermal relaxation, readout error etc. It would be beneficial if the authors can provide analysis of their generalization bounds for one of the above noise models, e.g. for depolarization noise.
>
> **Reply:**  Thank you for raising this important point. In response, we have incorporated the effect of local depolarizing noise acting after each quantum gate in the data re-uploading model. This noise introduces a noise-dependent term into the exponential component of our generalization bound. When the noise level is zero, our bound naturally reduces to the noiseless version. The related results are added in the Appendix.
>
> **Minor issues:**
>
> **Reply:**
> - We have double-checked the definition of CX gate, it is correct. The confusing point may be the sign is a direct sum.
>
> - We agree with you that QNN should comprise of three things: ansatz, observable and classical optimizer. We have revised correspondingly.
>
> - We have revised the Fig.2 and thanks for your careful review.
>
> - Thanks for your comments.

---

> ### Author Response · Authors · 2024-11-20
> **Questions 1-4**
>
> **Question 1:** Can your proof techniques be used for proving generalization bounds for data re-uploading QNNs where input-scaling parameters are also trainable? That is, the output function of a data re-uploading QNN is $f(\theta,\lambda,x,M)$, where are input-scaling parameters used when data is being re-uploaded and are trainable, e.g. data encoding operation is $U(x,\lambda)$ instead of $U(x)$ . In your current analysis, $\lambda = 1$ and is not trainable. I ask this because the function $f(\theta,\lambda,x,M)$ is more general and recovers your definition of $f(\theta,x,M)$ and can in principle do a bit more than your data reuploading model.
>
> **Reply:** Thanks for your thoughtful suggestion. We want to clarify that this case you mentioned is included in our framework.
> Specifically, for any encoding circuit $U(x,\lambda)$, it can be rewritten in the form $U(x,\lambda)=\prod_{j}U_j(\lambda)V_j(x)$.
> When combined with the original trainable circuit $U(\theta)$, this reconstructs a new data re-uploading model, which demonstrates that the re-uploading models defined in our paper are general enough to include such cases. We hope this explanation clarifies our approach and its applicability.
>
> **Question 2:** Does your definition of stability hold also for memorization algorithm[5]--a learning algorithm which always outputs the function $f(x_i,\theta_s, M) = -1$, except that it alters $f$ in a small number of locations to fit the training set?
>
> **Reply:** Thank you for this insightful question. To our understanding, stability characterizes the inherent properties of an algorithm itself. Our definition of stability focuses on quantifying the sensitivity of a model to perturbations in the training set, which aligns well with standard learning algorithms that aim for generalization. For such memorization algorithms, our stability definition may still holds in principle, as it would capture the localized instability introduced by these targeted alterations. However, it remains unclear how the stability measure would behave in all cases. Further investigation is needed to refine stability measures in such scenarios, particularly to address the trade-offs between memorization and generalization.
>
> **Question 3:** Can you provide any numerical evidence on the how tight is your analysis of generalization bound?
>
> **Reply:** We have to admit that it is challenge to calculate the theoretical generalization bound. In our current work, we rely on test loss as a proxy to estimate the generalization error, a commonly accepted approach within the community. Our experiments are designed to demonstrate that the generalization behavior observed aligns with the theoretical predictions derived in Corollary 4, providing evidence that the bound captures the expected trends in QNN performance.
>
> **Question 4:** What properties (or hyperparameters of QNNs) and operations (regularization, weight decay, gradient clipping etc.) induce stability?
>
> **Reply:** Thank you for your question. The stability of a learning algorithm primarily arises from the inherent randomness in the SGD process. In the context of studying algorithmic stability and the generalization of QNN models trained with SGD, hyperparameters such as the number of data re-uploading layers, the number of trainable parameters, and the learning rate explicitly influence the generalization error bounds. These factors serve as critical indicators of the model’s generalization performance and stability.
> This is also the motivation behind our title of `optimizer-dependent generalization'.

---

> ### Author Response · Authors · 2024-11-20
> **Question 5-7**
>
> **Question 5:** DNNs with random initialization tend to behave like random Gaussian processes (in limit of many neurons)[6] which allows for their characterization of their trainability and generalization. However, this is not the case for randomly initialized QNNs as they instead generally form Wishart processes[7] which give rise to barren plateau phenomenon and exponentially many local poor minima. That being said, does your generalization bound result help to mitigate these issues and if yes, how?
>
> **Reply:** This is a very insightful question, and we believe our results draw parallel conclusions with BP and local minima phenomenon, in the sense they both offer insights into the capability of QNNs.
> Here is our reasoning: The overall capability of QNNs can typically be characterized by three aspects: expressibility, trainability, and generalization. In data re-uploading model, it has been shown that as the number of layers increases, the expressibility of the model also increases. However, the growth in expressibility unavoidably leads to overfitting the training datasets, resulting in a high generalization gap, which is demonstrated in both our work and other studies. This overfitting implies that the model may not perform well on unseen data. Conversely, studies also point out that higher expressibility, achieved by adding more re-uploading layers, leads to smaller cost gradients. This indicates that the trainability of QNNs decreases as the number of layers increases. We believe that both results contribute valuable insights toward a broader understanding of QNNs, but from different perspectives. The paper you referenced is excellent, and we have cited it in the revised version.
>
> **Question 6:** Why is there a significant difference between the generalization error reported in Fig. 3 with settings [L=8 (purple or violet) with $\eta=0.01$] and Fig. 4 with settings [L=8 with $\eta=0.01$ (red)]. The generalization error should be more or less similar which is not the case if I look at the plots. Can you please explain this phenomenon?
>
> **Reply:** Thanks for your question. In our experiments, the initial parameters of the QNNs were randomly initialized for each independent training run. This stochastic initialization process can lead to variations in the training dynamics and, consequently, in the resulting generalization performance. While the same hyperparameter setting were used, different initial conditions can create slight variations in the optimization trajectory, let alone the randomness of SGD, leading to deviations in the final generalization error. In response to your feedback, we have revised Figure 2 by conducting additional simulations. The updated figures should now exhibit more consistent results.
>
> **Question 7:** In the numerical experiments to support your theoretical claims, why do you use parametrized quantum circuits (PQCs) that are not universal and classically simulable? This completely defeats the purpose of using QML, no? Because at the end, we want a quantum model (specified by a PQC) which is expressive (for the domain), trainable, noise robust and can not be classically simulable--holy grail of QML.
>
> **Reply:** Thanks for your question. Our primary goal is to develop a theoretical framework for understanding the interplay between generalization and classical optimizer aspects of QNNs, as outlined in Corollary 4. Since the theoretical derivations are independent of specific ansatz structures, we use such PQC to validate our theoretical insights. We acknowledge that achieving a QML model with properties such as expressivity, trainability, noise robustness, and classical non-simulability is the ultimate goal of the field. However, our current focus is on establishing and validating theoretical insights rather than demonstrating a specific advantage of QML over classical methods. We hope this explanation clarifies our approach and addresses your concerns.

---

> ### Author Response · Authors · 2024-11-26
> **Concluding remark**
>
> Thank you for your thoughtful and constructive feedback! We have carefully revised our manuscript to address your comments and strengthen our presentation. In particular, we have incorporated an analysis of the generalization bound in the presence of depolarizing noise. The changes incorporate your valuable suggestions while maintaining the clarity and rigor of our arguments. We believe these revisions have substantially improved the paper and hope they address your concerns adequately.

---

### Meta-Review · Area_Chair_CeUK · 2024-12-20

**Metareview:**

The paper explores the generalization performance of QNNs with a data re-uploading structure through the framework of algorithmic stability. The authors derive generalization error bounds using the uniform stability of the stochastic gradient descent algorithm and attempt to establish connections between QNNs and quantum combs. Simulation results provide partial validation of the theoretical findings. The submission received mixed reviews with varying scores and confidence levels. While reviewers acknowledged the significance of the topic and its potential insights, concerns about the technical contributions and the relevance of quantum combs limited the paper's overall impact.

**Additional Comments On Reviewer Discussion:**

The authors and reviewers engaged in discussions about the theoretical contributions, the practical significance of the generalization bounds, the role of quantum combs in their derivation, and the limited scope of the numerical experiments. During the rebuttal period, the authors provided clarifications, made revisions, and added experiments with a broader range of hyperparameter settings. They also included detailed dataset information and incorporated a discussion on the impact of depolarizing noise on the generalization bounds. Despite these efforts, two key issues remained unresolved after the discussion: 1. The technical contributions were considered limited, with substantial overlap with existing classical methods; 2. Questions persisted regarding the necessity of quantum combs in deriving the bounds, as the derivation did not clearly leverage the unique properties of this formalism.

---

### Decision · Program_Chairs · 2025-01-22

Reject